# Spatiotemporal dynamics of 53BP1 dimer recruitment to a DNA double strand break

Jieqiong Lou[1,2], David G. Priest[1,2], Ashleigh Solano[1,2], Adèle Kerjouan[2] & Elizabeth Hinde [1,2 ✉]

Tumor suppressor p53-binding protein 1 (53BP1) is a DNA repair protein essential for the detection, assessment, and resolution of DNA double strand breaks (DSBs). The presence of a DSB is signaled to 53BP1 via a local histone modification cascade that triggers the binding of 53BP1 dimers to chromatin flanking this type of lesion. While biochemical studies have established that 53BP1 exists as a dimer, it has never been shown in a living cell when or where 53BP1 dimerizes upon recruitment to a DSB site, or upon arrival at this nuclear location, how the DSB histone code to which 53BP1 dimers bind regulates retention and self-association into higher-order oligomers. Thus, here in live-cell nuclear architecture we quantify the spatiotemporal dynamics of 53BP1 oligomerization during a DSB DNA damage response by coupling fluorescence fluctuation spectroscopy (FFS) with the DSB inducible via AsiSI cell system (DIvA). From adopting this multiplexed approach, we find that preformed 53BP1 dimers relocate from the nucleoplasm to DSB sites, where consecutive recognition of ubiquitinated lysine 15 of histone 2A (H2AK15ub) and di-methylated lysine 20 of histone 4 (H4K20me2), leads to the assembly of 53BP1 oligomers and a mature 53BP1 foci structure.

[1] School of Physics, University of Melbourne, Melbourne, Victoria, Australia. [2] Department of Biochemistry and Molecular Biology, Bio21 Institute, University of Melbourne, Melbourne, Victoria, Australia. ✉email: elizabeth.hinde@unimelb.edu.au

DNA double-strand breaks (DSBs) represent a serious threat to genomic integrity. A single DSB left unrepaired can lead to mutations that promote chromosomal rearrangements, loss of an entire chromosome arm and ultimately cell death or oncogenic transformation[1,2]. To ensure cellular homeostasis, it is critical that this type of genomic lesion is detected with high fidelity and resolved accurately. To do just this, a cellular surveillance system termed the DNA damage response (DDR) has evolved and it resolves DSBs by one of two main DNA repair pathways—homologous recombination (HR) or non-homologous end joining (NHEJ)[3–5]. A key factor in a DSB's DDR is TP53-binding protein 1 (53BP1)[1,4]. 53BP1 is a large (1972 aa, ~214 kDa) DNA repair factor that is recruited to DSBs to regulate DNA repair pathway choice by blocking DNA end resection (part of HR) and promoting DSB repair by NHEJ[6–10]. Biochemically, it has been demonstrated that DSBs elicit a local histone modification cascade that triggers 53BP1 recruitment and binding of 53BP1 dimers to chromatin flanking a DSB[11,12]. However, the spatiotemporal dynamics that underpin this mechanism of 53BP1 dimer recruitment to a DNA DSB within the nuclear architecture of a living cell remains unknown.

53BP1 recognition of a DNA DSB starts with Ataxia-telangiectasia-mutated (ATM) kinase phosphorylating serine 139 of histone variant 2AX (γH2AX) within proximal chromatin[13–16] and a series of histone ubiquitination events performed by RING finger 8 (RNF8) and RNF168, which includes lysine 15 of core histone 2A (H2AK15ub)[11,17–22]. Although the biophysical mechanism by which 53BP1 molecules arrive at a chromatin-signaled DSB has not been directly observed within an intact cell nucleus, it is known from cryogenic electron microscopy that dimers of 53BP1 directly bind H2AK15ub-containing nucleosomes via the 53BP1 ubiquitylation-dependent recruitment motif and this interaction requires simultaneous engagement of the 53BP1 tandem tudor domain with dimethylated lysine 20 of histone 4 (H4K20me2)—a mark present throughout the genome[23]. It has also been shown by quantitative chemical proteomics that in addition to H2AK15Ub and H4K20me2, 53BP1 directly interacts with γH2AX via the 53BP1 BRCT domain in specific contexts such as late repairing foci[24].

53BP1 dimerization and self-association into higher-order oligomers is mediated by the 53BP1 oligomerization domain[25], which has been shown to promote 53BP1 recruitment to DSBs[26,27] and self-assembly of 53BP1 into phase-separated condensates[28]. Although it is known from co-immunoprecipitation experiments that 53BP1 dimers exist in the nucleoplasm independent of DNA damage signaling[25], it has never been shown in a living cell when or where 53BP1 dimers are formed during the DNA DSB recruitment phase. Furthermore, upon arrival at a DNA DSB, it is unknown how the different components of the DSB histone code (e.g., H2AK15Ub, H4K20me2, and γH2AX) regulate 53BP1 dimer retention or oligomer formation. Thus, here we set out to quantify the spatiotemporal dynamics of 53BP1 self-association during a DSB DDR and test whether the DSB histone code differentially regulates 53BP1 dimer recruitment, retention, and oligomer formation in the context of live-cell nuclear architecture. To do so we couple fluorescence fluctuation spectroscopy (FFS)[29,30] of fluorescently tagged constructs of 53BP1 with the DSB inducible via AsiSI cell system (DIvA)[31,32].

FFS describes a family of methods that statistically analyse fluctuations in fluorescence intensity within live-cell microscopy data to extract single molecule dynamics[29,30]. DIvA cells are a stable cell line generated in the human U2OS background harbouring a 4-hydroxytamoxifen (4OHT)-inducible AsiSI restriction enzyme, which allows for induction of approximately 100 site-specific DSBs in the genome upon 4OHT treatment[31,32]. Using the DIvA system, we designed two different types of FFS-

based experiments aimed at characterizing 53BP1 dimer localization and transport dynamics during DSB repair. The first experiment is based on Number and Brightness (NB) analysis[33–35], an FFS method that can extract the amplitude spectrum of eGFP-53BP1 fluorescence fluctuations recorded in a frame scan acquisition and convert this into a spatial map of the oligomeric state of 53BP1 throughout the nucleoplasm. The second experiment is based on cross-pair correlation function (pCF) analysis[36–38], an FFS method that can compare the temporal spectrum of eGFP-53BP1 fluorescence fluctuations with spatially distinct mKate2-53BP1 fluorescence fluctuations along a line scan and quantify 53BP1 dimer mobility in or across different chromatin environments.

From application of NB and cross-pCF to DIvA cells expressing different fluorescent constructs of 53BP1, and independent validation of this FFS-based strategy by fluorescence anisotropy imaging microscopy (FAIM) of homo-Förster resonance energy transfer (FRET)[39–41], we find the DSB histone code to regulate a spatiotemporal redistribution in 53BP1 oligomerization. Specifically, upon DSB induction a population of 53BP1 dimer that is present independent of DDR signaling is recruited to DSB sites. Upon arrival at these nuclear locations, H2AK15Ub mediates efficient 53BP1 dimer loading onto a DSB, while consecutive engagement with H4K20me2 enables this population of 53BP1 dimer to be retained. Collectively, it is this stepwise interaction that leads to the assembly of 53BP1 oligomers and mature 53BP1 foci.

## Results

### FFS reports the spatial distribution of eGFP-53BP1 oligomer formation in the nuclei of live DIvA cells.
Here we first establish a NB analysis workflow to spatially map the oligomeric state of 53BP1 within an FFS experiment and then quantify eGFP-53BP1 self-association in the nucleus of a live DIvA cell. To do so we perform NB FFS experiments in live DIvA transiently transfected with enhanced green fluorescent protein (eGFP) tagged to wild-type 53BP1 (eGFP-53BP1) vs. 53BP1 mutants that either have a diminished capacity to dimerize (eGFP-53BP1$^{YY1258,1259AA}$) or are constitutively oligomeric (eGFP-GCA-53BP1)[26]. An NB FFS experiment is a time series of confocal frames (Fig. 1a) that has been temporally optimized to capture fluctuations in eGFP-53BP1 fluorescence intensity within a diffraction limited pixel (Fig. 1b), and which upon application of a moment-based NB analysis (detailed in Methods), reports the apparent brightness of eGFP-53B1 in each pixel of the selected frame. To translate FFS detected values of eGFP-53BP1's apparent brightness into oligomeric state, NB analysis requires the apparent brightness of an eGFP monomer to be calibrated. Thus first we acquired an NB FFS experiment in the nucleus of a live DIvA cell transfected with free eGFP (Fig. 1c, d), and then from calculation of its mean apparent brightness (Fig. 1e, f), defined a series of brightness windows to detect and spatially map monomers, dimers or higher-order oligomers in eGFP-53BP1 NB FFS measurements.

From acquisition of NB FFS experiments in the nuclei of live DIvA cells transfected with eGFP-53BP1 vs. eGFP-53BP1$^{YY1258,1259AA}$ (negative control) and eGFP-GCA-53BP1 (positive control) (Fig. 1g, h), we find the eGFP-calibrated brightness windows (Fig. 1e) to accurately detect (Fig. 1i) and spatially map (Fig. 1j) 53BP1 dimer formation (green pixels), depletion (teal pixels), and promotion into higher-order oligomers (red pixels). In particular, we find from quantification of multiple nuclei that wild-type eGFP-53BP1 exists as both a dimer (26 ± 4%) and an oligomer (2 ± 1%) throughout the nucleoplasm, and as predicted by biochemical studies, this fraction undergoing self-association is reduced by introduction of the YYAA mutation

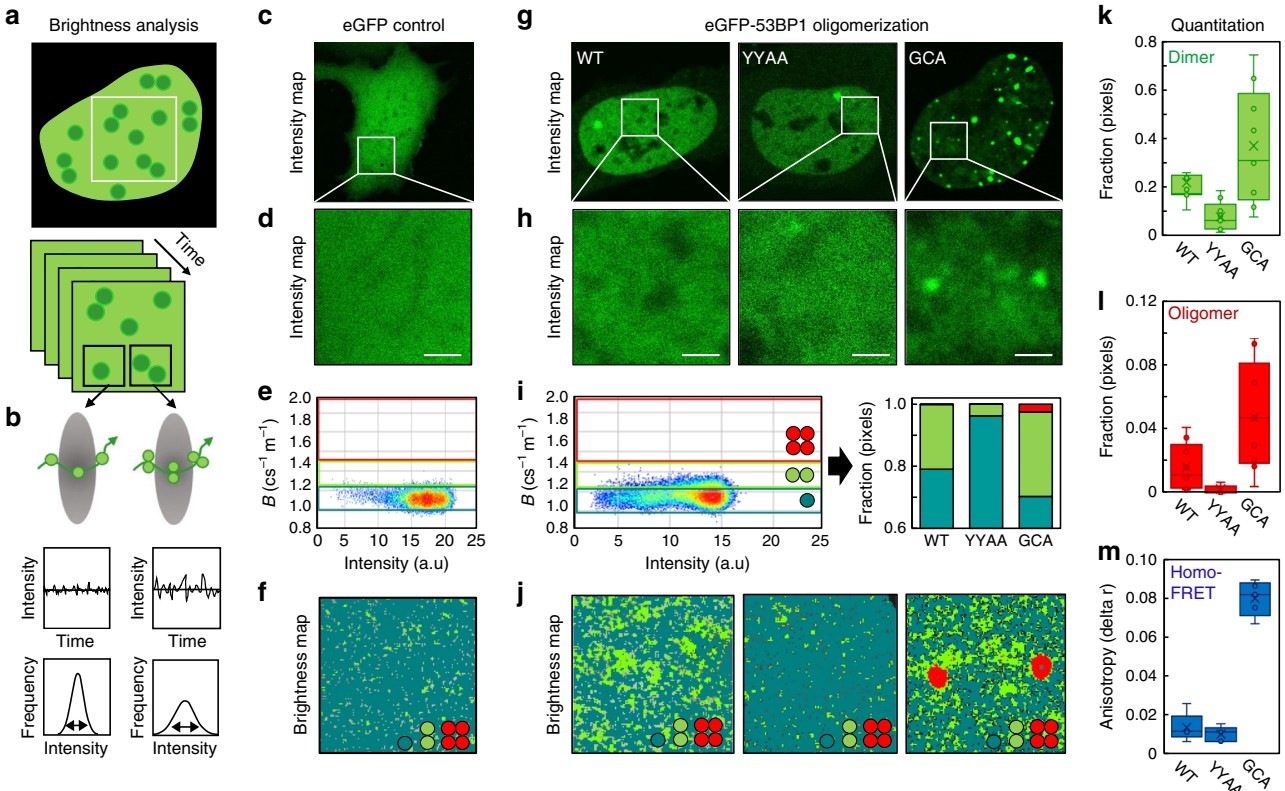

**Fig. 1 NB analysis reveals 53BP1 to exist as a dimer throughout the nucleoplasm. a** Schematic of a Number and brightness (NB) frame scan acquisition to measure eGFP-53BP1 oligomerization in a live DIvA cell. **b** Schematic of how eGFP-53BP1 monomer vs. oligomer diffusion through a diffraction limited point spread function of a frame scan pixel gives rise to a fluorescence fluctuation exhibiting a low vs. high variance with respect to the mean (definition of apparent brightness and the parameter that reports the oligomeric state of eGFP-53BP1). **c, d** Intensity image of a DIvA cell expressing eGFP (monomer calibration) (**c**) and the region of interest (ROI) from which an NB data acquisition was recorded (**d**). **e** Intensity vs. brightness scatterplot of the NB data acquisition presented in **d** reports the monomeric brightness of eGFP and enables extrapolation of brightness windows to detect eGFP-53BP1 dimer and oligomer formation. **f** Brightness map of the eGFP NB data acquisition in **d** pseudo-colored according to the brightness windows defined in **e**. **g, h** Intensity images of a DIvA cell expressing eGFP-53BP1 (WT), eGFP-53BP1^YY1258,1259AA (YYAA) and eGFP-GCA-53BP1(GCA) (**g**), and the ROI from which each NB data acquisition was recorded (**h**). **i** Intensity vs. brightness scatterplot of the eGFP-53BP1 NB data acquisition presented in **h** with the calibrated brightness windows superimposed (left) and quantification of the fractional contribution of 53BP1 monomer (teal), dimer (green), and oligomer (red) in WT vs. YYAA and GCA (right). **j** Brightness maps of the NB data acquisitions presented in **h** pseudo-colored according to the brightness windows defined in **i** spatially map 53BP1 monomer (teal), dimer (green), and oligomer (red) localization in WT vs. YYAA and GCA. **k, l** NB quantification of the fraction of eGFP-53BP1 dimer (**k**) vs. oligomer (**l**) present in WT vs. YYAA and GCA across multiple cells ($N = 10$ cells, two biological replicates). **m** Homo-FRET quantification of eGFP-53BP1 dimer and or oligomer formation in WT vs. YYAA and GCA across multiple cells ($N = 5$ cells, two biological replicates). Box and whisker plots in **k–m** show the minimum, maximum, sample median, and first vs. third quartiles. Scale bars, 2 μm.

and increased by the GCA construct (Fig. 1k, l). This result was verified by FAIM (Supplementary Fig. 1), where measurement of homo-FRET—an alternative readout of protein-protein interaction—recapitulated the shifts in 53BP1 self-association observed by NB (Fig. 1m). Thus, our NB analysis workflow does report bona fide eGFP-53BP1 self-association and under basal conditions we detect nuclear wide eGFP dimer and higher-order oligomer formation.

**NB analysis reveals preformed 53BP1 dimers inside the nucleoplasm to assemble into higher-order oligomers at DSB foci during the DDR.** To study whether and how eGFP-53BP1 self-association spatiotemporally redistributes during recruitment to DNA DSBs located throughout live-cell nuclear architecture, we next coupled our NB workflow of analysis (Fig. 1) with the capacity of the DIvA cell line to induce multiple site-specific DSBs after 4OHT treatment[31,32]. Immunofluorescence (IF) for γH2AX in fixed DIvA cells following 4OHT treatment confirmed (1) multiple DSB foci to form across the genome within 1 h of addition that were localized outside of HP1α foci (i.e., within

euchromatin) (Supplementary Fig. 2a, b)[42–44] and, importantly, for our live-cell study, (2) eGFP-53BP1 to co-localize at newly formed DSB foci (Fig. 2a, b and Supplementary Fig. 2b). We therefore acquired eGFP-53BP1 NB FFS data before and at 30 min intervals after 4OHT treatment (Fig. 2c), a time course that enabled observation of up to ten DSBs being formed throughout the nucleoplasm during the early DDR (Supplementary Fig. 2c, d). From NB analysis of eGFP-53BP1 dynamics across this time course, we derived a sequence of brightness maps pseudo-colored according to 53BP1 oligomeric state (Fig. 2d and Supplementary Fig. 3a). As can be seen from comparison of eGFP-53BP1 localization (Fig. 2c) and oligomerization (Fig. 2d), we find that with increasing time after 4OHT treatment there is a loss of 53BP1 dimer (green pixels) from the nucleoplasm that leaves behind a pool of monomeric 53BP1 (teal pixels), and this occurs in parallel with higher-order oligomer formation (red pixel) at the increasing number of 4OHT-induced DSB foci.

To quantify this DSB-induced spatiotemporal rearrangement in 53BP1 oligomer localization across multiple nuclei (Supplementary Fig. 3b, c), we next calculated the fraction of pixels assigned as

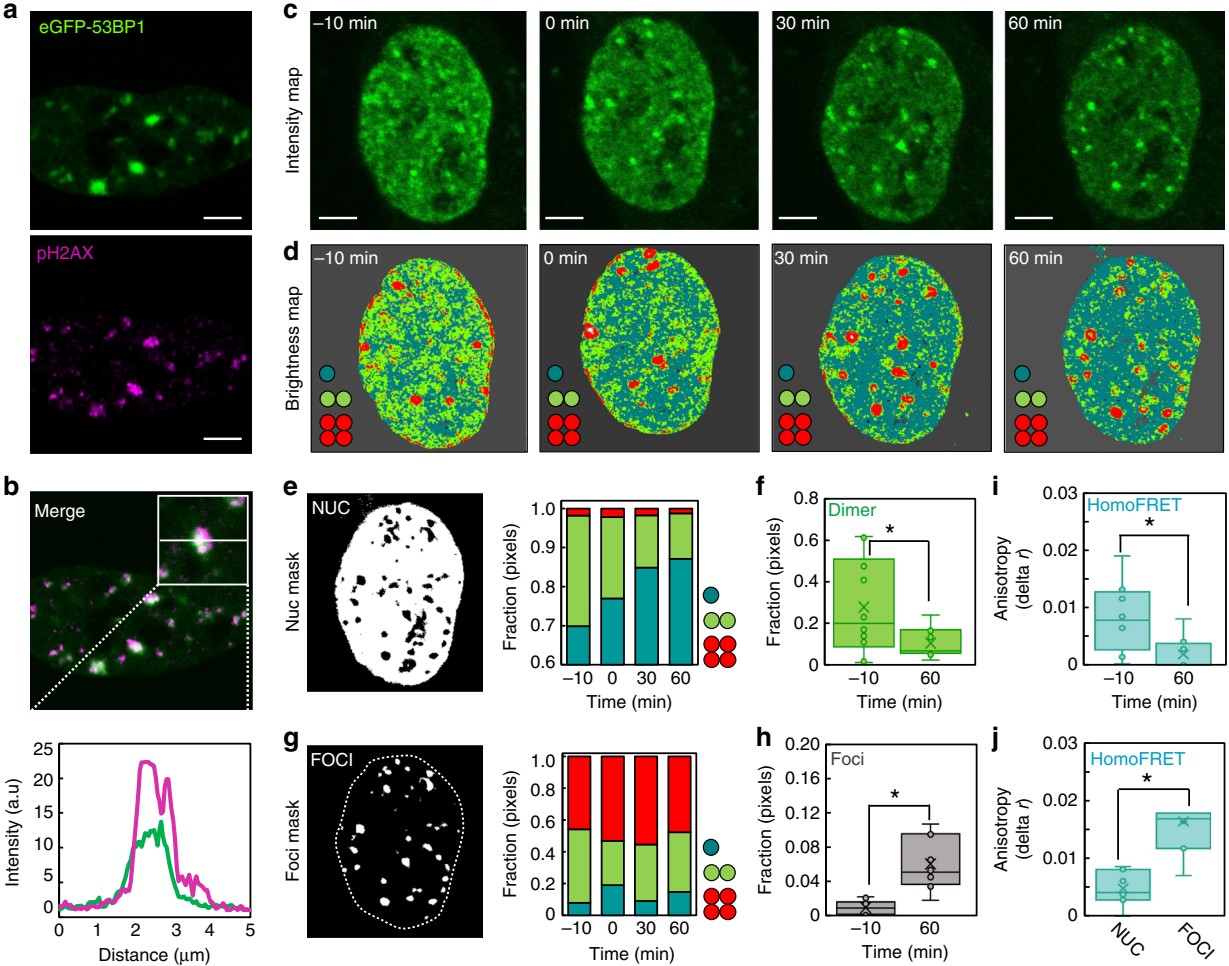

**Fig. 2 DSB induction initiates a nuclear wide relocalization of 53BP1 dimers to DSB sites. a, b** Co-localization of eGFP-53BP1 with γH2AX immunofluorescence confirms 53BP1 foci correlate with DNA DSBs in live DIvA cells (**a**) and enrich these nuclear locations (**b**). **c** Intensity images of NB FFS measurements acquired in a single DIvA cell expressing eGFP-53BP1 before and at different time points after 4OHT induction of DNA DSBs. **d** Brightness maps of the NB data acquisitions presented in **c** pseudo-colored according to the brightness windows defined in Fig. 1i. Corresponding intensity vs. brightness scatter plots are presented in Supplementary Fig. 3a and additional NB FFS data acquired at – 10 min and 60 min 4OHT treatment in Supplementary Fig. 3b, c. **e** Intensity mask highlighting the nucleoplasm (Nuc) (left panel) and quantification of the fractional contribution of 53BP1 monomer (teal), dimer (green) and oligomer (red) within this region of interest before and at different time points after 4OHT induction of DNA DSBs (right panel). **f** NB quantification of the fraction of 53BP1 dimer present in the nucleoplasm before and 60 min after 4OHT induction of DNA DSBs ($N = 10$ cells, two biological replicates). **g** Intensity mask highlighting the foci (left panel) and quantification of the fractional contribution of 53BP1 monomer (teal), dimer (green) and oligomer (red) within this region of interest before and at different time points after 4OHT induction of DNA DSBs (right panel). **h** NB quantification of the fraction of the nucleus occupied by 53BP1 foci before and 60 min after 4OHT induction of DNA DSBs ($N = 10$ cells, two biological replicates). **i, j**. Homo-FRET quantification of eGFP-53BP1 dimer and or oligomer formation in the nucleoplasm of DIvA cells before vs. after 4OHT treatment (**i**) and in the nucleoplasm vs. foci of DIvA cells after 4OHT treatment (**j**) ($N = 8$ cells, two biological replicates). Box and whisker plots in **f, h–j** show the minimum, maximum, sample median, and first vs. third quartiles. *$P < 0.05$ (unpaired *t*-test). Scale bars, 5 μm.

monomer, dimer and oligomer inside the nucleoplasm (Fig. 2e, f) vs. at DSB foci (Fig. 2g, h) as a function of time across multiple cells. This analysis confirmed that concomitant with significant loss of 53BP1 dimer from the nucleoplasm (Fig. 2f), 53BP1 is assembled into a steady state population of higher-order oligomer at the increasingly numerous DSB foci (Fig. 2h). Collectively, this result suggests that upon DSB induction, preformed 53BP1 dimers in the nucleoplasm are recruited to DSB lesions where they assemble into higher-order oligomers. FAIM of eGFP-53BP1 homo-FRET inside the nucleoplasm (Fig. 2i) vs. at DSB foci (Fig. 2j) verified this NB result. Also, importantly, this NB result was maintained upon selective knockdown of endogenous 53BP1 (Supplementary Fig. 4).

**Cross-pCF analysis tracks 53BP1 dimer recruitment within the nucleoplasm and retention at DSB foci during the DDR.** To directly track 53BP1 dimer recruitment and retention during DSB foci formation, we next performed cross-pCF analysis on FFS data acquired in DIvA cells co-transfected with eGFP-53BP1 and mKate2-53BP1. By labeling 53BP1 with two spectrally distinct fluorescent proteins (Fig. 3a) and acquisition of a two-channel confocal line scan across a 4OHT-induced DSB foci (Fig. 3b), we obtain FFS data (Fig. 3c) where the cross-pCF analysis can extract the following: (1) the fraction of eGFP-53BP1 diffusing as a complex with mKate2-53BP1 (i.e., 53BP1 dimers) within the nucleoplasm and (2) the time it takes this population of 53BP1 dimer to diffuse onto or off a DSB foci (Fig. 3d). For cross-pCF

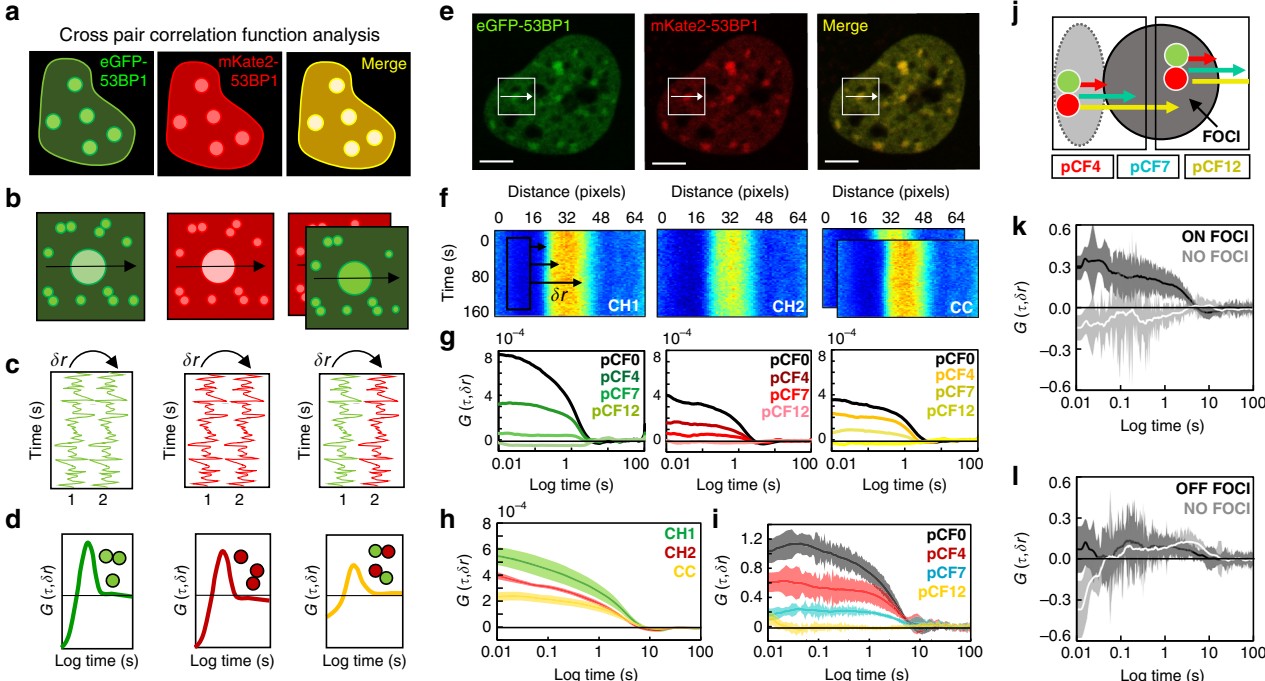

**Fig. 3 Cross pCF analysis tracks 53BP1 dimer recruitment and retention at DSBs. a**, **b**. Schematic of how co-labeling 53BP1 with eGFP and mKate2 (**a**) alongside acquisition of a two-channel line scan (**b**) enables 53BP1 dimers to be tracked by cross-pair correlation function (pCF) analysis. **c**, **d** Schematic of how the fluorescence fluctuations that result (**c**) can be spatially cross-correlated within each channel to track eGFP-53BP1 vs. mKate2-53BP1 transport or spatially cross-correlated between two channels (**d**) to track eGFP-53BP1 molecules in complex with mKate2-53BP1. **e** Confocal image of a DIvA cell nucleus co-transfected with eGFP-53BP1 and mKate2-53BP1 that has been treated with 4OHT. **f** Intensity carpet of a two-channel line scan acquired across a 53BP1 foci from the cell selected in **e**, in the green (CH1), red (CH2), and cross-correlated channels (CC) (schematic of the first 10 pixels cross-correlated at $\delta r = 0$, 4, 7, and 12 pixels (i.e., pCF0, pCF4, pCF7, and pCF12) is superimposed over CH1). **g** Overlay of the pCF0, pCF4, pCF7, and pCF12 profiles derived in the green (CH1) vs. red (CH2) channels of the two-channel line scan presented in **f** and the corresponding cross-pCF profiles (yellow) (pCF0 = 80 nm (pixel diameter), pCF4 = 320 nm, pCF7 = 560 nm, and pCF12 = 980 nm). **h** Overlay of cross pCF0 analysis of 53BP1 dimer mobility (yellow) with pCF0 analysis of total 53BP1 mobility (green, red) to extract the fraction of 53BP1 dimer present in the nucleoplasm ($N = 5$ cells, shading indicates SEM). **i** Overlay of cross-pCF4, cross-pCF7, and pCF12 analysis of 53BP1 dimer transport from the nucleoplasm onto a DSB foci normalized with respect to pCF0 to compare transit efficiency ($N = 5$ cells, shading indicates SEM). **j** Schematic of the spatial evolution of 53BP1 dimer transport onto vs. off a DSB foci (pCF7 reaches outside the point spread function). **k**, **l** Cross-pCF7 analysis of 53BP1 dimer translocation onto a DSB foci (**k**) vs. off this structure (**l**) that is normalized with respect to cross pCF0 ($N = 5$ cells, shading indicates SEM). Scale bars, 5 μm.

analysis to be effective at tracking 53BP1 dimer diffusion with respect to a DSB foci, however, we must first optimize the distance ($\delta r$) at which we spatially cross correlate the two different colored 53BP1 fluorescence fluctuations. Thus, in co-transfected DIvA cells treated with 4OHT for 60 min (Fig. 3e) we (1) acquired two-channel line scan data across DSB foci (Fig. 3f) and (2) performed pCF (green and red profiles) vs. cross-pCF analysis (yellow profile) of 53BP1 mobility within each pixel ($\delta r = 0$), as well as over three different spatial scales ($\delta r = 4$, 7, and 12 pixels) (Fig. 3g) to establish the fraction of 53BP1 dimer present and the condition in which we track this population onto a DSB foci.

From pCF analysis of eGFP-53BP1 and mKate2-53BP1 local mobility (monomers and dimers) in the nucleoplasm at $\delta r = 0$ (pCF0), and comparison of this analysis with the cross pCF0 profile of 53BP1 dimers only, we find the fraction of 53BP1 dimer present to be $31 \pm 4\%$ (Fig. 3h) after correction for spectral bleed through (Supplementary Fig. 5a–c). From cross-pCF analysis of this 53BP1 dimer fraction's transport onto a DSB foci at a distance of $\delta r = 4$, 7, and 12 pixels (pCF4 = 320 nm, pCF7 = 560 nm, and pCF12 = 980 nm), we find the spatial evolution of this transport to be effectively captured with cross-pCF4 to pCF7 (30–50% efficiency) and not beyond this distance range at cross pCF12 (0% efficiency) (Fig. 3i). Given that the dimensions of our line scan results in cross-pCF4 favoring local mobility (Fig. 3j), we proceeded with cross-pCF7 that favours 53BP1 dimer transport

onto a DSB foci and compared this transit with 53BP1 dimer transport off this structure (Fig. 3k, l and Supplementary Fig. 5d–i). This analysis revealed that (1) the presence of a DSB regulates the recruitment of 53BP1 dimer onto a DSB, as in the absence of a DSB, no 53BP1 dimer translocation is detected (Fig. 3k) and, (2) although significant 53BP1 dimer translocation is detected onto a DSB, no 53BP1 translocation is detected off this structure (Fig. 3l). Thus, collectively these results suggest that 53BP1 dimers are recruited to DSBs and upon arrival accumulate at this location, because 53BP1 dimers do not translocate off a DSB.

**The DNA DSB histone code differentially regulates the spatiotemporal distribution of 53BP1 oligomerization during the DDR.** NB (Fig. 2) and cross-pCF analysis (Fig. 3) have so far shown that preformed 53BP1 dimers are recruited to DSBs, and upon arrival, they immobilize as well as self-associate into higher-order oligomers. Next, to dissect the role of the DSB histone code that 53BP1 binds, in regulation of 53BP1 self-association during this recruitment mechanism, we performed NB FFS measurements on 53BP1 mutants that inhibit recognition of H4K20me2 (eGFP-53BP1[D1521R])[12], H2AK15ub (eGFP-53BP1[L1619A])[11,23], and γH2AX (eGFP-53BP1[K1814M])[24]. In the literature, it is reported that H4K20me2 and H2AK15ub are critical for DSB recruitment of 53BP1[11,12], whereas γH2AX plays a secondary role in 53BP1 DSB

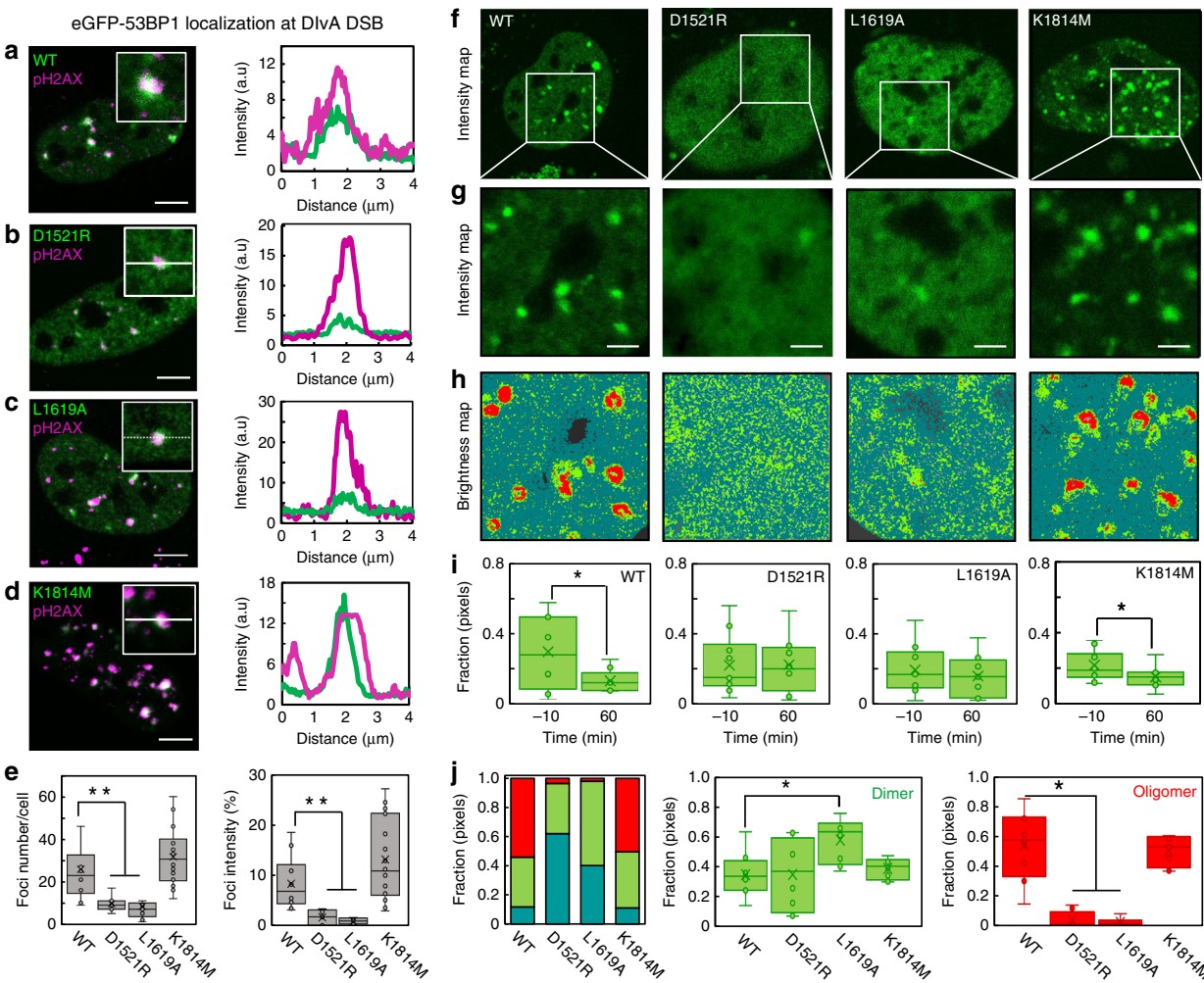

**Fig. 4 The DSB histone code regulates a spatiotemporal redistribution in 53BP1 oligomerization. a–d** Co-localization of eGFP-53BP1 (WT) (**a**), eGFP-53BP1[D1521R] (D1521R) (**b**), eGFP-53BP1[L1619A] (L1619A) (**c**), and eGFP-53BP1[K1814M] (K1814M) (**d**) with γH2AX immunofluorescence in a single cell (left panels) and across a single-foci (right panels). Scale bars, 5 μm. **e** Quantification of foci number (left panel) and foci intensity (right panel) per cell at 60 min following 4OHT DSB induction (N > 30 cells). **f, g** Intensity images of a DIvA cell expressing WT, D1521R, L1619A, and K1814M at 60 min following 4OHT DSB induction (**f**) and the region of interest from which an NB frame scan acquisition was recorded in each case (**g**). Scale bars, 2 μm. **h** Brightness maps of the NB data acquisitions presented in **f, g** pseudo-colored according to the brightness windows defined in Fig. 1i. Corresponding intensity vs. brightness scatter plots are presented in Supplementary Fig. 6a, as well as additional NB FFS data acquired 60 min after 4OHT treatment of these different 53BP1 constructs (Supplementary Fig. 6b, c). **i** NB quantification of the fraction of WT, D1521R, L1619A, and K1814M dimer in the nucleoplasm before and 60 min after 4OHT induction of DNA DSBs (N = 10 cells, two biological replicates). **j** NB quantification of the fractional contribution of monomer, dimer, and oligomer in WT, D1521R, L1619A, and K1814M foci 60 min after 4OHT induction of DNA DSBs (left) (N = 10 cells, two biological experiments). Box and whisker plots in **e**, **i**, **j** show the minimum, maximum, sample median, and first vs. third quartiles. *P < 0.05, **P < 0.01 (unpaired t-test).

foci formation[24]. Thus, in DIvA cells transfected with eGFP-53BP1 vs. eGFP-53BP1[D1521R], eGFP-53BP1[L1619A], or eGFP-53BP1[K1814M], we first performed γH2AX IF 60 min after 4OHT treatment (Fig. 4a–d), to verify the reported effects of each histone mark on 53BP1 recruitment to DSBs in DIvA cells (Fig. 4e). Then we acquired NB FFS data on the different eGFP-tagged 53BP1 mutants before and 60 min after 4OHT treatment (Fig. 4f–h and Supplementary Fig. 6), to uncover the impact each histone mark has on the following: (1) the spatiotemporal redistribution of 53BP1 dimers upon DSB induction (Fig. 4i) and (2) self-association of 53BP1 dimers into oligomers during foci formation (Fig. 4j).

In agreement with the literature we find that 60 min after 4OHT treatment, the number of DSBs enriched with eGFP-53BP1[D1521R] and eGFP-53BP1[L1619A] in the DIvA cell system is significantly less than eGFP-53BP1 or GFP-53BP1[K1814M] (Fig. 4e, left). Along this line, from NB analysis of eGFP-53BP1[D1521R] and

eGFP-53BP1[L1619A], we find that unlike eGFP-53BP1[K1814M], their significantly diminished capacity to recognize and accumulate at DSBs (Fig. 4e, right), also disrupts the spatiotemporal redistribution in oligomeric state previously observed for wild-type 53BP1 (Fig. 4h). In particular, although recognition of H4K20me2 and H2AK15ub is not required for 53BP1 dimer formation, these two histone marks are critical for the significant relocation of 53BP1 dimer from the nucleoplasm to DSBs (Fig. 4i) and maintenance of 53BP1 oligomeric foci composition (Fig. 4j, left panel). Specifically, upon arrival at the few DSBs that do lead to eGFP-53BP1[D1521R] or eGFP-53BP1[L1619A] foci formation, we detect the following: (1) dimer accumulation in the absence of H2AK15ub recognition (Fig. 4j, middle panel) and (2) no oligomer formation in the absence of either H4K20me2 or H2AK15ub recognition (Fig. 4j, right panel). Thus, collectively this result demonstrates that the DSB histone code is central to 53BP1 dimer recruitment,

retention, and oligomer formation at DSB sites, and 53BP1 dimer recognition of H4K20me2 vs. H2AK15ub might be consecutive.

**H2AK15ub is important for 53BP1 dimer recruitment to a DNA DSB and H4K20me2 is key for 53BP1 dimer retention.** NB showed that the individual components of the DSB histone code differentially regulate the spatiotemporal redistribution of 53BP1 dimers from the nucleoplasm to DSB foci (Fig. 4i) and interaction with H2AK15ub vs. H4K20me2 may be a sequential process rather than a simultaneous event (Fig. 4j). To further dissect this result, we next quantified the impact H4K20me2, H2AK15ub and γH2AX have on 53BP1 dimer loading onto DIvA DSBs vs. retention at these nuclear locations, by cross-pCF analysis of line scan FFS data acquired in DIvA cells co-transfected with eGFP and mKate2 labeled 53BP1, 53BP1$^{D1521R}$, 53BP1$^{L1619A}$, and 53BP1$^{K1814M}$ (Fig. 5a–d). This analysis revealed that while the spatial evolution of wild-type 53BP1 dimer transport onto DSB foci was effectively captured with an efficiency of 30% (extracted from cross-pCF7 amplitude) and without delay (extracted from cross-pCF7 timescale) (Fig. 5e), the 53BP1 mutants altered this efficiency and the timing upon which 53BP1 dimers are loaded onto DSBs, as well as the net population of 53BP1 dimers retained (Fig. 5f–h).

Specifically, from cross-pCF7 analysis of 53BP1$^{D1521R}$ (Fig. 5f), we find that in the absence H4K20me2 recognition, 53BP1 dimer transport onto DSB foci is delayed by ~20 ms and the net population of 53BP1 dimer retained at the DSB site is reduced to an efficiency of 10%, because significant 53BP1 dimer transport off the DSB is enabled. From cross-pCF7 analysis of 53BP1$^{L1619A}$ (Fig. 5g), we find that in the absence of H2AK15ub recognition, 53BP1 dimer transport onto a DSB foci is delayed by ~100 ms and the net population of 53BP1 dimer retained at the DSB site is reduced to an efficiency of 10%, because 53BP1 dimer transport onto the DSB site is significantly disrupted. Finally, from cross-pCF7 analysis of 53BP1$^{K1814M}$ (Fig. 5h), we find that in the absence of γH2AX recognition, 53BP1 dimer transport onto DSB foci is unperturbed temporally and the net population of 53BP1 dimer retained at the DSB site is captured at an efficiency of 20%. Thus, collectively, these results alongside NB analysis of 53BP1$^{D1521R}$, 53BP1$^{L1619A}$ and 53BP1$^{K1814M}$ (Fig. 4f–j) demonstrate that while all three histone marks are important for efficient 53BP1 dimer recruitment to a DSB, consecutive recognition of H4K20me2 and H2AK15ub upon arrival at these nuclear locations is critical for 53BP1 dimer assembly into higher-order oligomers and foci formation. In particular, 53BP1 dimer interaction with H2AK15ub is required for efficient loading onto

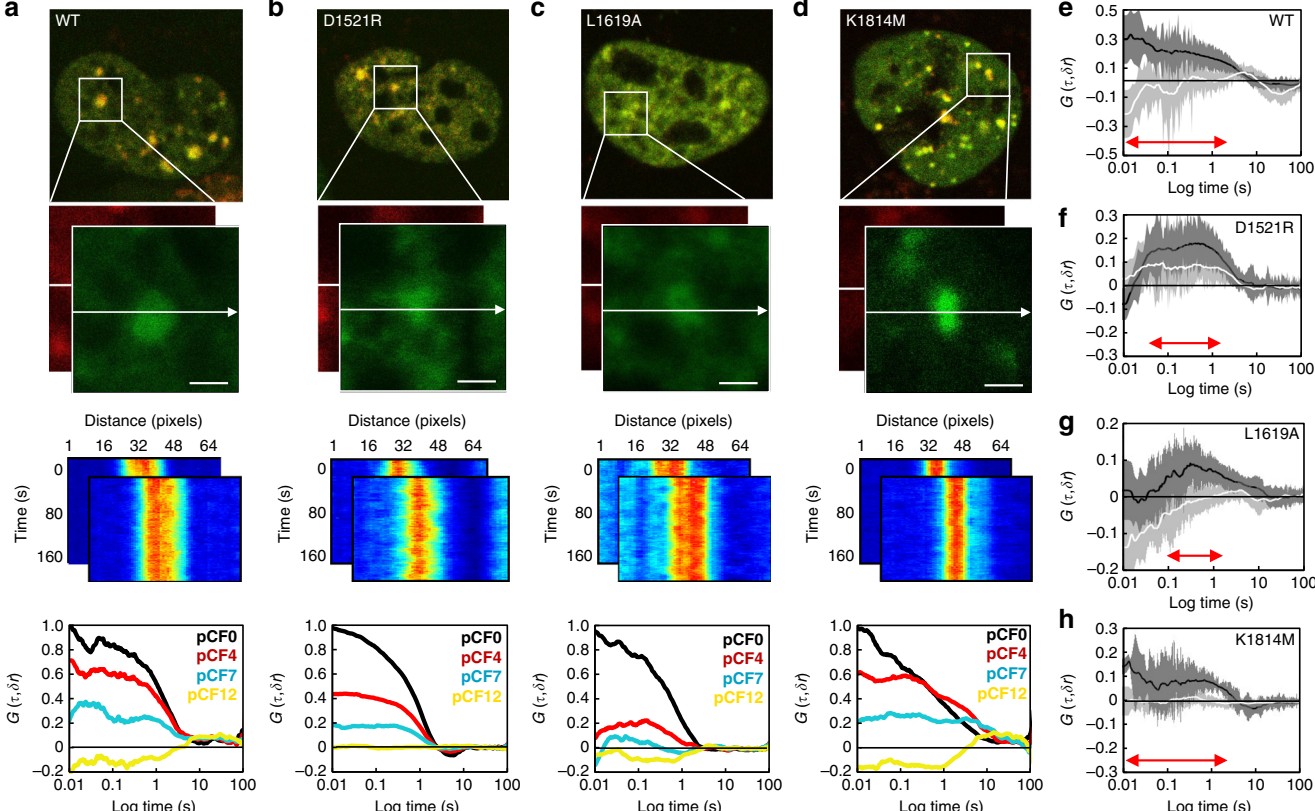

**Fig. 5 The DSB histone code differentially regulates 53BP1 dimer transportation and retention onto a DSB. a–d** Two color merged confocal image of a DIvA cell nucleus co-transfected with eGFP and mKate2 tagged 53BP1 (WT) (**a**), 53BP1$^{D1521R}$ (D1521R) (**b**), 53BP1$^{L1619A}$ (L1619A) (**c**), and 53BP1$^{K1814M}$ (K1814M) (**d**), which has been treated with 4OHT for 60 min (top row) and the region of interest selected for acquisition of a two-channel line scan (second row). Intensity carpet of the two-channel line scan acquired across a selected 53BP1 WT, D1521R, L1619A and K1814M foci in the green and red channels (third row), as well as cross-pCF analysis of the spatial evolution of 53BP1 WT, D1521R, L1619A, and K1814M dimer transport onto this structure at a distance of $\delta r = 0$, 4, 7, and 12 pixels (i.e., pCF0, pCF4, pCF7, and pCF12) (bottom row). In agreement with cross-pCF analysis of 53BP1 WT in Fig. 3, cross-pCF7 was optimal for tracking D1521R, L1619A, and K1814M dimer translocation onto the DSB foci (cross-pCF4 tracks local mobility while pCF12 does not detect 53BP1 dimer translocation with significant efficiency). **e–h** Cross-pCF7 analysis of 53BP1 WT (**e**), D1521R (**f**), L1619A (**g**), and K1814M (**h**) dimer translocation onto a DSB foci (black) vs. off this nuclear structure (white) ($N = 10$ cells, two biological replicates). Shading indicates SEM and red arrows indicate in each case the distribution of time delays detected. All cross-pCF curves in **a–d** and **e–h** are normalized with respect to cross pCF0. Scale bars, 2 μm.

the DSB site and 53BP1 dimer engagement with H4K20me2 is required for immobilization. Knockdown of RNF8/168 and chemical inhibition of SUV4-20h1/2 that ubiquitinate H2AK15 and methylate H4K20, respectively, verified these histone marks are key for 53BP1 transport (Supplementary Fig. 7).

## Discussion

In this study we coupled a multiplexed approach to FFS with the DIvA cell system, to investigate the spatiotemporal dynamics of 53BP1 dimer recruitment, retention, and oligomer formation at DSBs in a living cell. In doing so, we detect a nuclear wide spatial redistribution in 53BP1 self-association that is differentially regulated by the histone modification cascade local to DSBs. To our knowledge, this is the first demonstration of where and when 53BP1 self-associates with respect to DSB induction in live-cell nuclear architecture.

In particular, from NB analysis of wild-type 53BP1 oligomer localization in DIvA cells we found in agreement with biochemical studies[25,26] that 53BP1 dimers exist throughout the nucleoplasm independent of DDR signaling (Fig. 1). Then, within 30–60 min of DSB induction, this nuclear population of 53BP1 dimer is spatially redistributed to DSB sites where they assemble into higher-order oligomers (Fig. 2). A cross-pCF analysis of 53BP1 dimer transport during this spatial redistribution revealed that 53BP1 dimer recruitment is specifically regulated by the presence of DSBs and their accumulation at these nuclear locations is due to an absence of 53BP1 dimer translocation off DSB sites on the timescale of our experiment (Fig. 3). We propose that it is the difference in the on vs. off rate of 53BP1 dimer translocation with respect to DSBs that facilitates 53BP1 dimer self-association into higher-order oligomers, and higher-order oligomer formation may serve to stabilize 53BP1 foci structure by bridging interactions across multiple nucleosomes nearby the DSB. Thus, in order to determine what regulates 53BP1 dimer recruitment vs. retention at DSBs, we next dissected the role of the multicomponent DSB histone code to which 53BP1 dimers bind in spatiotemporally coordinating this DDR dependent redistribution in 53BP1 oligomerization.

From IF of γH2AX (Fig. 4a–e) and FFS analysis of 53BP1 mutants that inhibit interaction with the DSB histone code (Figs. 4f–j and 5), we found in agreement with the literature[11,12,23,24] that H4K20me2 and H2AK15ub recognition is essential for significant relocation of 53BP1 dimers from the nucleoplasm to DIvA DSBs and oligomer assembly at these nuclear locations, whereas γH2AX plays a facilitatory role in 53BP1 foci formation. Intriguingly, from cross-pCF analysis of 53BP1 mutants with defects in key histone modification recognition (Fig. 5), we found that while H2AK15ub is critical for the timely and efficient loading of 53BP1 dimers onto a DIvA DSB,

H4K20me2 is important for subsequent 53BP1 dimer retention at these nuclear locations. This result alongside NB (Fig. 4f–j), which detected 53BP1 dimer accumulation at DSBs in the absence of reading H2AK15ub but not H4K20me2, suggests 53BP1 dimer interaction with these two histone marks is a consecutive event. In particular, 53BP1 dimers engage sequentially with H2AK15ub and H4K20me2 via a 'capture' and 'lock' mechanism (Fig. 6a, b) that stimulates oligomer assembly and 53BP1 foci formation (Fig. 6c, d). Collectively, these findings are in keeping with the recent observation that the inner core of a 53BP1 foci that ultimately evolves into a liquid phase droplet, is initiated by cooperative interaction with DSB histone marks[28].

In conclusion, here we demonstrate an FFS-based extraction of 53BP1 dimer dynamics in live DIvA cells that uncovers a nuclear wide spatial redistribution in 53BP1 oligomerization, which is coordinated by the DSB DDR and underpinned by consecutive interactions with the DSB histone code. We envisage that the detected enrichment of DSBs with 53BP1 dimers that self-associate into oligomers plays a critical role in maintaining 53BP1 foci structure, and via interaction with neighbouring nucleosomes, could function to modulate the compaction status of the surrounding chromatin environment. Thus, future studies will be dedicated towards investigating the impact of 53BP1 oligomerization on DSB site chromatin structure, for example, by measurement of FRET between fluorescent histones[45]. Given the recent report that sub-micron domains within 53BP1 foci regulate chromatin topology[46], it will also be important to place this dynamic picture of 53BP1 self-association in the context of a super-resolved image of 53BP1 foci formation[47].

## Methods

**Cloning**. 53BP1 mutants YY1258, 1259AA, D1521R, and L1619A were generated using mutagenesis via Gibson Assembly[48]. Briefly, pcDNA5-FRT/TO-eGFP-53BP1 (Addgene 60813) was digested with AgeI-HF and BamI-HF (NEB) followed by Antarctic Phosphatase treatment (NEB: M0289). The vector fragment was then gel extracted (NEB Monarch gel extraction kit). Overlapping PCR primers containing the desired mutant codons paired with common PCR primers with ~30 bp overhangs for assembly into the digested eGFP-53BP1 vector were used in two separate PCR reactions (NEB Q5 polymerase). PCR products were then gel extracted and assembled with the vector (~1 ng/100 bp of DNA), transformed into chemically competent *Escherichia coli* DH5α cells (Thermo Fisher 18265017), and plated onto selective Lysogeny broth agar media. The 53BP1 mutant K1814M and small interfering RNA (siRNA)-resistant 53BP1 constructs on the other hand, were generated using gBlocks (IDT) flanking the desired changed base pairs. The digested vector and gBlock were assembled by Gibson Assembly with a 1 : 3 molar ratio of vector to gBlock. eGFP in the eGFP-53BP1 vectors was replaced with mKate2 using Gibson Assembly with a PCR product (mKate2_f, 5′-tccggactc-tagcgtttaaacttaagcttggtaccatggtgagcgagctgattaagg-3′; mKate2_r, 5′-aactgacttccag-tagggtccattggcgcgcctctgtgccccagtttgctagg-3′), which had compatible overhangs and cloning it into Acc651/AscI digested 53BP1 vector. All constructs and mutants were confirmed by Sanger sequencing (AGRF, Melbourne).

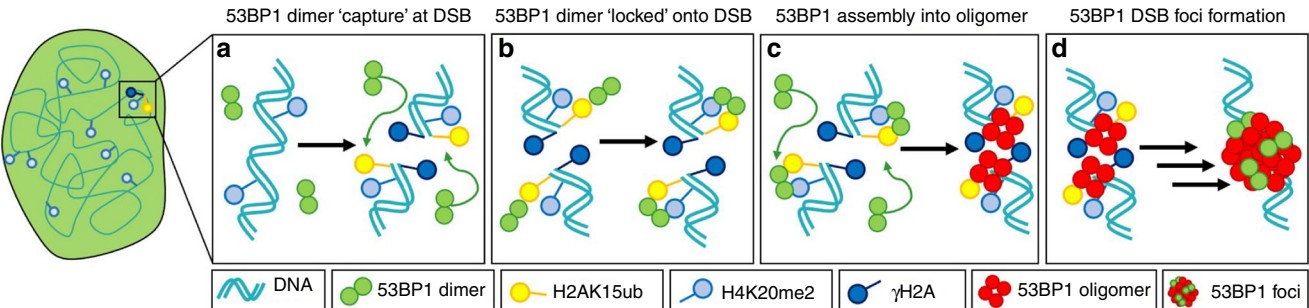

**Fig. 6 Schematic of the 'capture' and 'lock' mechanism of 53BP1 foci formation. a–d** FFS analysis of 53BP1 dynamics in live DIvA cells revealed that upon DSB induction H2AK15ub mediates efficient 53BP1 dimer capture onto a DSB site (**a**), H4K20me2 serves a lock function for 53BP1 dimer retention at this nuclear location (**b**), and it is this cooperative interaction with H2AK15ub and H4K20me2, which initiates 53BP1 oligomerization (**c**) and 53BP1 foci formation (**d**).

**Cell culture, transient transfection, and IF**. DIvA cells (originally provided by Gaëlle Legube, LBCMCP, CNRS, Toulouse, France) were grown in Dulbecco's modified Eagle's medium (Lonza) supplemented with 10% bovine growth serum (Gibco), 1× Pen-Strep (Lonza), and 1 µg/ml puromycin (Thermo Fisher Scientific) at 37 °C in 5% $CO_2$. For live-cell microscopy experiments, the DIvA cells were plated 24 h before experiments onto 35 mm glass bottom dishes and transiently transfected or co-transfected with the following plasmids via use of Lipofectamine 3000 according to the manufacturer's protocol: (1) eGFP (Addgene, #54767), eGFP-53BP1 (Addgene, #60813), eGFP-53BP1$^{YY1257,1258AA}$, eGFP-GCA-53BP1 (kindly provided by Professor Thanos D. Halazonetis, University of Geneva, Switzerland), eGFP-53BP1$^{D1521R}$, eGFP-53BP1$^{L1619A}$, eGFP-53BP1$^{K1814M}$, mKate2-53BP1, mKate2-53BP1$^{D1521R}$, mKate2-53BP1$^{L1619A}$, and mKate2-53BP1$^{K1814M}$. For IF against γH2A.X (S139) (Catalog number 9718S, Cell Signaling) and 53BP1 (Catalog number 4937S, Cell Signaling), cells were fixed with 4% paraformaldehyde for 15 min, permeabilized with 1 mg/ml Triton X-100 for 15 min at room temperature, and blocked with 1% bovine serum albumin, each in a phosphate-buffered saline buffer. Primary antibody (1 : 200 dilution) was incubated overnight at 4 °C. Secondary antibody (Catalog number A21244, Invitrogen 1 : 1000) was incubated for 1 h at room temperature. For DNA staining, cells were incubated with 1 µM Hoechst 33342 for 10 min at room temperature.

**Transient knockdown of protein expression and siRNA-resistant eGFP-53BP1 expression**. All siRNA-mediated knockdown of RNF8, RNF168 and 53BP1 was achieved using Lipofectamine RNAiMAX (Thermo Fisher) mediated transfection according to the manufacturer's protocol: 25 pmol of siRNA duplexes per 0.25 million cells. 53BP1 siRNA sequence was as follows 5′-AGAACGAGGAGACGG UAAUAGUGGG-3′. To generate siRNA-resistant 53BP1 expression constructs, A231G, A234G, and A237C silent point mutations were introduced into the 53BP1 cDNA[49]. RNF8 and RNF168 siRNA were as follows, respectively: RNF8 5′-ATG GTAAACTGTACGCTA-3′, 5′-TTCAGAATAGCGTACAGT-3′; RNF168 5′-GAA AGCTAAGCTAAGCATTGATA-3′, 5′-TTGTTAATATCAATGCTT-3′ (from IDT). The siRNA-resistant eGFP-53BP1 construct was transiently transfected via use of Lipofectamine 3000 according to the manufacturer's protocol at 24 h post siRNA transfection. Microscopy experiments were conducted at 48 h after DIvA cells being transfected with siRNA duplexes and IF was used to test 53BP1 knockdown efficiency at the same time.

**Confocal laser scanning microscopy**. All microscopy measurements were performed on an Olympus FV3000 laser scanning microscope coupled to an ISS A320 Fast FLIM box for fluorescence fluctuation data acquisition. A ×60 water-immersion objective 1.2 NA was used for all experiments and the cells were imaged at 37° in 5% $CO_2$. For single channel NB FFS measurements the eGFP-tagged plasmids were excited by a solid-state laser diode operating at 488 nm and the resulting fluorescence signal directed through a 405/488/561 dichroic mirror to a photomultiplier detector (H7422P-40 of Hamamatsu) fitted with an eGFP 500/25 nm bandwidth filter. For the dual channel cross-pCF FFS measurements the eGFP and mKate2 plasmids were excited by solid-state laser diodes operating at 488 nm and 561 nm, respectively, and the resulting signal was directed through a 405/488/ 561 dichroic mirror to two internal GaAsP photomultiplier detectors set to collect 500–540 nm and 600–700 nm, respectively. For the FAIM measurements of homo-FRET the eGFP-tagged plasmids were excited by a solid-state laser diode operating at 488 nm and the resulting fluorescence signal directed through a 405/488/561 dichroic mirror as well as a thin film polarizer cube, which split the parallel and perpendicular signal with respect to the excitation light into two photomultiplier detectors (H7422P-40 of Hamamatsu). For the IF measurements Hoechst 33342 and Alexa Fluorophore 647 (AF647) were excited by solid-state laser diodes operating at 405 nm and 640 nm, respectively, and the resulting signal was directed through a 405/488/561/640 dichroic mirror to two internal GaAsP photomultiplier detectors set to collect 430–470 nm and 650–750 nm, respectively.

**Microscopy data acquisition**. NB FFS measurements of the different eGFP-tagged 53BP1 constructs involved selecting a DIvA cell exhibiting a sufficiently low expression level to observe fluctuations in eGFP fluorescence intensity[29,30] and then selecting a 10.6 µm region of interest (ROI) within that DIvA cell's nucleus (Figs. 1 and 4) (or an ROI that contained the entire cell nucleus (Fig. 2)), which for a 256 × 256 pixel frame resulted in a pixel size of 41 nm (an oversampling of the point spread function of our diffraction limited acquisition). A frame scan acquisition (n = 100 frames) was then acquired with the pixel dwell time set to 12.5 µs, which resulted in a line time of 4.313 ms and a frame time of 1.108 s. Cross-pCF FFS measurements of the different eGFP and mKate2 tagged 53BP1 constructs involved selecting a DIvA cell exhibiting low and comparable co-expression of the two fluorescent constructs and then selecting a 5.3 µm line across the middle of a 4OHT-induced DSB foci (Figs. 3 and 5), which for a 64 × 1 pixel line resulted in a pixel size of 83 nm (which again oversamples the point spread function). A two-channel line scan acquisition (n = 100,000 lines) was then acquired with the pixel dwell time set to 8 µs, which resulted in a line time of 1.624 ms (our sampling frequency). FAIM of the different eGFP-tagged 53BP1 constructs employed the same scan settings as NB (Figs. 1 and 2).

**NB analysis**. The brightness of a fluorescently tagged protein is a readout of that protein's oligomeric state that can be extracted by a moment-based NB analysis of an FFS frame scan acquisition[33–35]. In brief, within each pixel of a frame scan we have an intensity fluctuation ($F(t)$) that has an average intensity $\langle F(t) \rangle$ (first moment) and a variance $(F(t) - \langle F(t) \rangle)^2$ (second moment). As defined in Eq. (1), the ratio of these two properties describes the apparent brightness ($B$) of the molecules that give rise to the intensity fluctuation.

$$B = \frac{(F(t) - \langle F(t) \rangle)^2}{\langle F(t) \rangle} \tag{1}$$

The true molecular brightness ($\varepsilon$) of the molecules is related to the measured apparent brightness ($B$) by $B = \varepsilon + 1$, where 1 is the brightness contribution of our photon counting detector. Calibration of the apparent brightness of monomeric eGFP ($B_{monomer} = 1.15$) enabled extrapolation of the expected apparent brightness of eGFP-53BP1 dimers ($B_{dimer} = 1.30$) and oligomers ($B_{oligomer} > 1.60$) (Fig. 1d–g), which in turn enabled definition of brightness cursors to extract (Fig. 1h, i) and spatially map (Fig. 1j) the fraction of pixels within a given frame scan acquisition that contain these different species. The fraction of eGFP-53BP1 dimer and eGFP-53BP1 oligomer (i.e., number of pixels assigned $B_{dimer}$ or $B_{oligomer}$) were used as parameters to quantify the degree of 53BP1 self-association detected across multiple cells (Fig. 1k, l). An intensity-based mask was used to quantify these parameters in the nucleoplasm vs. within foci. Artefact due to cell movement or photobleaching were subtracted from acquired intensity fluctuations via use of a moving average algorithm. All brightness calculations were carried out in SimFCS from the Laboratory for Fluorescence Dynamics (www.lfd.uci.edu).

**Homo-FRET analysis**. FRET between identical fluorophores is termed homo-FRET and it is a readout of protein-protein interaction (e.g., dimer formation) that can be monitored by FAIM[39–41]. In brief, homo-FRET induces a depolarization to a fluorescence emission that can be detected as a reduction in fluorescence anisotropy, and as defined by Eq. (2), anisotropy ($r$) is the intensity-corrected difference between the emission parallel ($I_\parallel$) and emission perpendicular ($I_\perp$) to the exciting vector.

$$r = \frac{I_\parallel - gI_\perp}{I_\parallel + g2I_\perp} \tag{2}$$

The g factor accounts for any systemic differences introduced by the optics or detectors into the parallel vs. perpendicular channel and this was calibrated in our system by measurement of fluorescein, a fast-rotating molecule whose emission polarization should be isotropic. Depolarization effects caused by use of a high numerical aperture objective were taken-into-account by incorporation of an additional correction factor into $r$ that was derived from referencing eGFP anisotropy values acquired with a 60X water objective to eGFP anisotropy values acquired with a ×10 air objective[50]. After application of this correction factor to FAIM measurements acquired in live DIvA cells expressing free eGFP we find that in our system the fluorescence anisotropy value of eGFP in the absence of homo-FRET ($r_{eGFP}$) is 0.32. In the presence of homo-FRET due to eGFP-53BP1 dimer or oligomer formation, the expectation is that $r_{eGFP}$ will be reduced. Thus, we express eGFP-53BP1 homo-FRET in terms of delta $r$, which is equal to the difference between $r_{eGFP}$ and $r_{eGFP-53BP1}$. An intensity-based mask was used to quantify these parameters in the nucleoplasm vs. within foci. All anisotropy and homo-FRET analysis of FAIM data acquisitions were carried out in a custom code written in Matlab.

**Cross-pCF analysis**. Cross-pCF analysis of spectrally and spatially distinct fluorescence fluctuations acquired along a two-channel FFS line scan acquisition can track the evolution of dimer transport[36–38]. In brief, in each pixel of a two-channel line scan we have an intensity fluctuation ($F(t)$), which we can format into two intensity carpets, where the x-coordinate in each carpet corresponds to the point along the line (pixel) in that channel and the y-coordinate corresponds to the time of acquisition. The carpet data format enables temporal cross correlation of pairs of intensity fluctuations separated by a set distance ($\delta r$) for every possible delay time ($\tau$) either (1) within a single channel, by application of the pCF function (defined in Eq. 3) that tracks all molecules present[36,51] or (2) between the two channels, by application of the cross-pCF function (defined in Eq. 4) that tracks only molecules in a complex[37,38]:

$$G(\tau, \delta r) = \frac{\langle F(t,0) \cdot F(t+\tau, \delta r) \rangle}{\langle F(t,0) \rangle \langle F(t, \delta r) \rangle} - 1 \tag{3}$$

$$G_{cross}(\tau, \delta r) = \frac{\langle F1(t,0) \cdot F2(t+\tau, \delta r) \rangle}{\langle F1(t,0) \rangle \langle F2(t, \delta r) \rangle} - 1 \tag{4}$$

The pCF and cross-pCF profiles that result from cross correlation of intensity fluctuations within or between channels, respectively, reports the time ($\tau$) it takes a population of molecules (or in the case of cross-pCF, only those molecules in a complex) to translocate a set distance ($\delta r$), and because the measurement is exquisitely local to a pair of points, the translocation time into or out of different environments along the line scan can be extracted with a spatial resolution of ~260 nm (radial axis of the point spread function (PSF)). Thus, given that the two-channel FFS line scan measurements presented here were acquired with a pixel size

of ~80 nm, the spatial evolution of 53BP1 transport was tracked by performing pCF and cross-pCF analysis at a distance of $\delta r = 0, 4, 7$, and 12 ($\delta r < 4$ pixels detects local mobility within a single observation volume vs. $\delta r > 4$ that detects translocation between two observation volumes). These four pCF distances (pCF0 = 80 nm, pCF4 = 320 nm, pCF7 = 560 nm, and pCF12 = 960 nm) enabled quantification of the fraction of 53BP1 dimer present in the nucleoplasm, the time it takes this fraction to diffuse onto a DSB vs. off this structure and the efficiency of these two transits at a selected optimal distance.

To quantify the fraction of 53BP1 dimer present in the nucleoplasm the amplitude of the pCF0 profile derived for eGFP-53BP1 mobility ($G0_{CH1}$) and mKate2-53BP1 mobility ($G0_{CH2}$) were compared with the amplitude of the cross pCF0 profile derived for eGFP-53BP1 in complex with mKate2 mobility ($G0_{CC}$) and the ratio was taken with the limiting channel (i.e., $G0_{CC}/G0_{CH1}$ if $G0_{CH1} < G0_{CH2}$ or $G0_{CC}/G0_{CH2}$ if $G0_{CH2} < G0_{CH1}$) (Supplementary Fig. 5a). A bleed through correction was then applied to this calculated fraction of dimer by subtraction of a cross correlation index ($G0_{BT}$)[52] that was derived from simulation of the experimentally measured percentage of eGFP signal detected in the mKate channel (Supplementary Fig. 5b, c). To quantify the time it takes the quantified fraction of 53BP1 dimer to diffuse from the nulceoplasm onto a DSB at cross-pCF4, pCF7, and pCF12, the pCF0 profile from which these transits begin was used to normalize each cross-pCF profile and the 'efficiency' parameter[53] is defined as the ratio between the peak amplitude of pCF4, pCF7, or pCF12 with pCF0. As described with respect to Fig. 3 we found cross-pCF7 to be optimal for tracking 53BP1 dimer translocation onto vs. off DSB foci, as cross-pCF4 tracks local mobility within a pixel and pCF12 did not detect 53BP1 dimer translocation with significant efficiency. Thus all cross-pCF7 profiles presented in Figs. 3 and 5 are normalized with respect to pCF0 in the nucleoplasm (Supplementary Fig. 5d–f) vs. within foci (Supplementary Fig. 5g–i) and retention efficiency is the difference in amplitude between normalized pCF7 profiles reporting 53BP1 translocation onto a DSB vs. off this structure. Artefact due to cell movement or cell bleaching were subtracted from acquired intensity fluctuations via use of a moving average algorithm. All pCF calculations were carried out in SimFCS from the Laboratory for Fluorescence Dynamics (www.lfd.uci.edu).

**IF quantification and foci counting**. For 53BP1 knockdown quantification, DIvA cell nuclei were identified by thresholding the corresponding Hoechst 33342 signal in dual channel IF confocal images by ImageJ. The coordinates of the identified nuclei were saved in the ImageJ ROI manager and then the mean 53BP1 intensity in the identified ROIs was calculated via use of the ImageJ 'measurement' function. The number and intensity of γH2AX, 53BP1 WT, and 53BP1 mutant foci were quantified in ImageJ. Confocal images were smoothed twice with the smooth function and foci were thresholded out by selecting the top ~5% pixels. Each identified ROI was counted as a foci and foci intensity was calculated by the sum intensity of all foci divided by sum intensity of the whole nucleus.

**Statistics and figure preparation**. Statistical analysis was performed by using GraphPad Prism software. Figures were prepared by use of Adobe Illustrator, MATLAB, and SimFCS.

**Reporting summary**. Further information on research design is available in the Nature Research Reporting Summary linked to this article.

## Data availability
The datasets generated during and/or analysed during the current study are available from the corresponding author on reasonable request.

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

## Acknowledgements

We thank Dr. Gaëlle Legube and Dr. Thomas Clouaire for providing the DSB inducible via AsiSI cell system (DIvA). We thank Professor Thanos D. Halazonetis for kindly providing 53BP1 mutant plasmids. We thank Professor Enrico Gratton for useful discussion on data analysis. We thank Jee Khor for useful discussion on sample preparation. J.L. is supported by an Australian National Health and Medical Research Council (NHMRC) project grant (APP1104461) and Albert Shimmins Research Continuity Funding. D.P. is supported by an Australian Research Council (ARC) discovery project (DP180101387). A.K. is supported by an Australian NHMRC project grant (APP1121907). E.H. is supported by an Australian NHMRC Career Development Fellowship (APP1124762) and the Jacob Haimson Beverly Mecklenburg Lectureship. We thank the Biological Optical Microscopy Platform, University of Melbourne, for enabling access to the Olympus FV3000 confocal laser scanning microscope.

## Author contributions

J.L. and E.H. conceived the study and designed experiments. J.L. conducted experimentation. D.P. and A.K. provided and created unique reagents. J.L., A.S., and E.H. analysed the data. J.L. and E.H. wrote the manuscript.

## Competing interests

The authors declare no competing interests.
