## [Peer Review File · Nature Communications]

Reviewers' comments:

Reviewer #1 (Remarks to the Author):

This is a review of "Stepwise 53BP1 foci assembly in response to DNA double strand break." In this manuscript, the homodimerization process of the 53BP1 following DSB induction is examined. To investigate this mechanism, the authors used coupling fluorescence fluctuation spectroscopy on DiVa cells. By adopting this technique, the authors also showed that the loading of 53BP1 dimers to the DNA breaks is mediated by H2AK15ub dependent as well as H4K20me20. The data presented in the manuscript is likely to be of considerable interest to the DNA repair community, but the current form of the manuscript is not suitable for publication to Nature Communications. Due to several major and minor shortcomings, this paper in its present form is not, in my opinion, suitable for publication, but it can be improved substantially by addressing the issues raised below

General point of criticism:

The paper in its present form is purely descriptive. It does not put the Authors' findings in a biological context, and offers no hypothesis as to where, when and most importantly how 53BP1 dimerize before DDR induction (although the authors start with the following sentence the discussion: To our knowledge this is the first demonstration where and when 53BP1 self-associates). No real hypothesis of the mechanism, meaning or functional importance is offered. Because of this reason the following questions must be addressed:

1. In DiVa cells the AsiSI creates over 200DSB, in contrast on figure 1-4 following 4OHT addition only 10-20 53BP1 foci could be observed following DSB induction. This at least should be addressed in order to exclude the possibility that following 4-OHT addition the ER translocation process is not sufficient. At least the co-localization of 53BP1-GFP and another DSB sensor (part of MRN complex or ATM) must be shown.
2. In the MS it is not clarified whether the 53BP1 loading measured in euchromatic or heterochromatic environment. I think it is an important point and the data need to be separated accordingly.
3. Why wasn't the 53BP1 homodimerization measured with fluorescent protein-protein interaction visualization or in fixed cells the proximity ligation assay would be good tool to demonstrate that 53BP1 mostly exist in an already dimerized form.
4. The authors use conventional microscopy with an advanced processing method to clarify their findings. The resolution of a conventional confocal microscopic method is approximately 200-300nm. This should be discussed in the text. Additionally, a recent report from Claudia and Jiri Lukas lab used showed mechanistical data on 53BP1 loading by using 3D-SIM. This has to be discussed in the manuscript. Additionally, I also suggest applying quantitative super resolution microscopy at least to show that the 53BP1 is associated to DSB in dimer form (Varga 2019 Nanoscale).
5. In the manuscript the pre-existing endogenous 53BP1 foci must be addressed or at least discussed (Figure 1e, Figure 2e, Figure 3a and e)
6. Figure 1g and h the authors conclude that the amount of monomer and dimer forms of 53BP1 are decreasing while oligomerisation could be observed (This result suggests that upon DSB induction pre-formed 53BP1 dimers in the nucleoplasm are recruited to DSB lesions and upon arrival assemble into higher order oligomers.) For me the data represented on figure 1h show the opposite. Additionally, Figure 1i is not discussed in the text.

Minor points:

1. On figure 2 4, 7- and 12-pixel distances are shown. I think it would be better if they would give the distance in nm resolution.
2. On figure 3 since the 53BP1 foci formation is affected the authors need to use control to show that the DSB induction is good (ATM-mCherry for example)
3. In order to show that H4K20me and H2aK15ub indeed affects the 53BP1 loading they should perform shRNA knockdown of these PTM writers (SUV4-20h1/2 and RNF8-RNF168 double KD) to

validate their results.

4. In the introduction the authors say that 53BP1 is the heart of DDR. I think it is an overstatement and has to be removed. I agree that this is an important player but not the middle of DDR.
5. The following statement also need to be modified: 53BP1 is rapidly recruited to DSBs. Since 53BP1 is considered as not an early factor this sentence needs to be re-phrased.
6. on page 5 last chapter it is written: the population of 53BP1 molecules being tracked was lost. What does it mean?

Reviewer #2 (Remarks to the Author):

The manuscript entitled "Stepwise 53BP1 foci assembly in response to DNA double strand break" by Lou and colleagues reports that 53BP1 foci are assembled at sites of DSBs by recruitment of pre-formed 53BP1 dimers via a H2AK15ub dependent mechanism, and upon arrival, H4K20me2 is critical for immobilization, while H2AK15ub stimulates formation of higher order oligomers that lead to a mature DSB repair focus. This conclusion is based entirely on the FFS technique and would be greatly strengthened, if similar conclusions could be reached by alternative methods. The authors confirm previous reports that binding of 53BP1 to H4K20me2 is required for recruitment of 53BP1 to foci, but surprisingly observe recruitment of the 53BP1-L1619A mutant, which is defective in H2AK15ub binding, to foci, which is contradictory to the literature. Also, the manuscript ignores reports that 53BP1 also interacts with phospho-H2AX-S139 (Kleiner et al., 2015), which limits the scope of the work. Given these concerns and additional issues listed below, I recommend that this work would be better suited for a specialist journal where space would permit proper explanation of the fluorescence fluctuation spectroscopy technique and addition of relevant control and complementary experiments.

Major concerns:

1. The authors do not report which fraction of the 53BP1 molecules in the cell are GFP-tagged. Western blotting should be used to confirm the level of endogenous 53BP1 compared to the GFP-53BP1 transgene.
2. The calibration of GFP-53BP1 to oligomeric state is not convincing. Foci of different oligomeric states should have relative fluorescence intensities of 1, 2, 3, 4, 5, etc, if the authors are truly detecting single molecules. The authors need to introduce a control where the oligomeric state is known with certainty e.g. a chromatin-bound protein fused to 1, 2, 3 and 4 molecules of GFP.
3. Introduction: the current knowledge on 53BP1 oligomeric states (monomers, dimers, tetramers) should be reviewed in more detail to put the study in context.
4. Fig. 1f-g: Have the images been adjusted for photobleaching? It looks like the overall fluorescence decreases with time.
5. Fig. 1f-g: The indication of oligomers as tetramers is misleading. If I understand the data correctly, there brighter simply contains a greater number of 53BP1 molecules above a certain threshold. The time-lapse quantification looks more like immobilization of diffuse nuclear 53BP1 at DSBs. What is the number of 53BP1 molecules in the brighter foci?
6. Fig 1h: N = 10 cells. This is a very low number of cells. How many biological replicates were analyzed?
7. Fig 1i: The fraction of pixels in foci does not reflect the example in Fig 1f, where it looks like the number of red pixels is similar at all time points.
8. Are the eGFP-53BP1 and mKate2-53BP1 constructs expressed at equal levels. Western blots should be provided.
9. The coexpression of eGFP-53BP1 and mKate2-53BP1 should reveal monomers that are either green or red, and dimers that are either half green and half red or entirely green or red (assuming no endogenous 53BP1 and equal amounts of eGFP-53BP1 and mKate2-53BP1 are expressed). Can

this be observed?

10. Fig 3: very few cells are analyzed and apparently only 1 biological replica? What is the variation in multiple biological replicates?

11. Fig 4d-f: The eGFP-53BP1-L1619A foci observed after DSB-induction have a different morphological appearance than WT foci. The authors should confirm that these foci colocalized with gamma-H2AX and are not inclusion bodies due to the L1619A mutation. Notably, other labs have reported that 53BP1-L1619A is defective for recruitment to DSB-induced foci (Fradet-Turcotte et al., 2013).

12. The assessment of monomeric and dimeric forms of 53BP1 should be complemented by alternative methods e.g. FLIM-FRET.

13. The authors propose that dimers and monomers have different on/off rates at foci. This should be supported by FRAP analysis.

Minor comments:

1. Fig S1a: There appears to be no gammaH2AX foci in the untreated sample and also no background in these cells. This is unlike what is normally reported in the literature, where a background of spontaneous foci are normally observed. Were the images subjected to the same contrast enhancement as the 4OHT induced cells?

2. Page 3: Fig 1e does not appear to show 15 min intervals according to the labels (-10, 0, 10, 30 min)?

3. Page 12: please reference the "previously published papers".

4. Page 5: I don't think the data justifies 4 significant digits (27.62%).

5. Page 7: it has not been formally shown that 53BP1 dimers "self-associate into higher order oligomers". It might be that multiple dimers bind to individual nucleosomes within a focus.

6. Page 7: what is the evidence that "under this condition only 53BP1 dimers were found to be tracked with respect to DSB foci"?

References:

Fradet-Turcotte, A., Canny, M.D., Escribano-Diaz, C., Orthwein, A., Leung, C.C., Huang, H., Landry, M.C., Kitevski-LeBlanc, J., Noordermeer, S.M., Sicheri, F., et al. (2013). 53BP1 is a reader of the DNA-damage-induced H2A Lys 15 ubiquitin mark. *Nature (London)* 499, 50-54.

Kleiner, R.E., Verma, P., Molloy, K.R., Chait, B.T., and Kapoor, T.M. (2015). Chemical proteomics reveals a gammaH2AX-53BP1 interaction in the DNA damage response. *Nat Chem Biol* 11, 807-814.

Reviewer #3 (Remarks to the Author):

In this manuscript, Lou et al. analyze in living cells the oligomeric state of 53BP1 in and out DNA repair foci using fluorescence fluctuations methods. Based on N&B and pair correlation data, the authors propose that, outside repair foci, 53BP1 mainly exists as dimer, these dimers thus forming higher oligomers upon relocation to repair foci. They also analyze the impact on the 53BP1 oligomerization status of two mutations impairing binding to specific histone marks and propose that H4K20me2 is important for immobilisation of 53BP1 dimers at repair foci while H2AK15ub stimulates formation of higher order oligomers.

The method presented in this manuscript is interesting and potentially applicable to other biological questions since it allows to assess directly in living cells protein oligomeric state as well as dynamic binding to DNA repair foci. The biological insights which are presented are valuable although relatively limited. In addition, I believe that additional experiments are necessary to more convincingly support the conclusions made by the authors.

I would advise the authors to address the following points :

1) The experiments are currently performed using cells over-expressing fluorescently tagged

53BP1. Thus, there is a possibility that 53BP1 dimers are composed of a tagged molecule as well as an untagged one, which would be detected as a monomer by N&B. The coexistence of these two versions of 53BP1 in the cells makes the interpretation of the present results somehow difficult. Either the size of the oligomers is underestimated by the presence of significant amount of untagged 53BP1 or the amount of endogenous 53BP1 is considered as neglectable compared to the quantity of tagged 53BP1 molecules. However, this last possibly supposes high levels of overexpression of GFP-53BP1, thus questioning biological relevance. I think that it would be important to perform the experiments presented in this manuscript using cells knocked-down or knocked-out for endogenous 53BP1. In addition, it would also be useful to control that the expression of GFP-53BP1 in these cells is comparable to endogenous level to avoid potential artifactual oligomerisation of GFP-53BP1 due to too high expression.

2) Fluorescence fluctuation methods usually require relatively low local concentration of molecules to be able to measure fluorescence fluctuations of sufficient amplitudes. This tends to question the ability to obtain accurate N&B results at bright spots such as DSB repair foci. The authors should demonstrate that they can perform N&B analysis at such high concentration of molecules. For example, they could perform N&B in nuclei with fluorescent spots obtained by the accumulation of tagged Lac repressor at Lac operator arrays. Since Lac repressor are known form tetramers (see e.g. Friedman et al., Science, 1995, doi: 10.1126/science.7792597) this could be used as a control to validate N&B analysis.

3) For the two mutants of 53BP1 that do not strongly accumulate at DNA repair foci, it would be important to co-express a second marker allowing to localize the DSBs. This would allow to make sure that pCF analysis is indeed performed along lines crossing DSBs.

4) For the two mutants of 53BP1, the authors compared single-color pCF curves with the results obtained by cross pCF for the wild-type 53BP1 (Fig 3j-k Fig 4j-k). It seems inappropriate to compare results obtained with two different approaches, one allowing to specifically look at dimers (cross pCF), while the other one probes both monomers and oligomers (single-color pCF). I would suggest that the authors perform cross pCF for the two 53BP1 mutants as they did for the wild-type version of 53BP1. Also, it would be useful to show single-color pCF for wild-type 53BP1 leaving the repair foci. Currently, the authors show by cross pCF that 53BP1 do not leave the foci as dimer but it would be interesting to know whether they could leave the foci as monomers.

5) In several instances in the manuscript, the authors give very few details regarding some data processing steps. For example, this is the case for the bleedthrough correction (Fig S3) or the calculation of the fraction of dimers (Fig 2i). It would be important to explain better how these data processing steps are performed.

6) The current version of the discussion is relatively short. I would find interesting to discuss more extensively the biological implications of the results presented in this manuscript. More specifically, the authors should discuss their results in the context of the recent publication by Kilic et al. (Kilic et al., EMBO J, 2019, doi: 10.15252/embj.2018101379) which shows that 53BP1 displays the propensity to undergo phase separation in the nucleoplasm.

Response to Referees Letter

Reviewer # 1 Remarks to the Author

This is a review of "Stepwise 53BP1 foci assembly in response to DNA double strand break." In this manuscript, the homodimerization process of the 53BP1 following DSB induction is examined. To investigate this mechanism, the authors used coupling fluorescence fluctuation spectroscopy on DiVA cells. By adopting this technique, the authors also showed that the loading of 53BP1 dimers to the DNA breaks is mediated by H2AK15ub dependent as well as H4K20me20. The data presented in the manuscript is likely to be of considerable interest to the DNA repair community, but the current form of the manuscript is not suitable for publication to Nature Communications. Due to several major and minor shortcomings, this paper in its present form is not, in my opinion, suitable for publication, but it can be improved substantially by addressing the issues raised below.

We thank Reviewer 1 for their overall feedback and in response we have now acquired new experimental data to address the major and minor concerns raised below. Incorporation of this data into a majorly revised manuscript has greatly strengthened our biological conclusions. Highlights based on Reviewer 1's suggestions include: (1) independent measurement of 53BP1 dimer-oligomer formation via homo-FRET (an alternative read out of protein-protein interaction) (detailed in new **Fig. 1** and **Fig. S1**), (2) quantitative characterisation of 53BP1 foci formation and the spatiotemporal kinetics of DSB induction in the DiVA cell line via γ H2AX immunofluorescence (new **Fig. S2** and **Fig. 4a-e**) and (3) confirmation that H2AK15ub / H4K20me2 are critical for the detected 53BP1 dimer transport dynamics by siRNA KD of RNF8/168 and chemical inhibition of SUV4-20h1/2 (new **Fig. S7**).

General point of criticism: The paper in its present form is purely descriptive. It does not put the Authors' findings in a biological context, and offers no hypothesis as to where, when and most importantly how 53BP1 dimerize before DDR induction (although the authors start with the following sentence the discussion: To our knowledge this is the first demonstration where and when 53BP1 self-associates). No real hypothesis of the mechanism, meaning or functional importance is offered. Because of this reason the following questions must be addressed:

Major concerns:

1. In DiVA cells the AsiSI creates over 200DSB, in contrast on figure 1-4 following 4OHT addition only 10-20 53BP1 foci could be observed following DSB induction. This at least should be addressed in order to exclude the possibility that following 4-OTH addition the ER translocation process is not sufficient. At least the co-localization of 53BP1-GFP and another DSB sensor (part of MRN complex or ATM) must be shown.

Yes, it is true that from genome-wide ChIP analyses of DiVA cells the AsiSI enzyme was found to generate approximately 200 site-specific DSBs across the genome (Massip et al. 2010 Cell Cycle). However, when imaging DiVA cells by live cell confocal microscopy, we and the original DiVA characterisation paper (Massip et al. 2010 Cell Cycle) only observe 10-20 53BP1 foci at any point in time after 4OHT treatment (now presented in **Fig. S2d**) for two main reasons: (1) 53BP1 foci are dynamic – the lifetime of a single DiVA DSB foci visualised by eGFP-53BP1 is on average around 30 minutes (now presented in **Fig. S2c**), and therefore, it is impossible to observe all 200 DSB at any given time point by live cell microscopy, and (2) we only image one x-y plane out of a 3-dimensional DiVA nucleus, therefore, there are many other out of focus foci in the other z planes. We now address this issue in the form of new supplementary figure (**Fig. S2**), where, as suggested by Reviewer 1 we also present quantification of 53BP1-GFP co-localisation with γ H2AX immunofluorescence (a DSB sensor) (**Fig. S2b**).

2. In the MS it is not classified whether the 53BP1 loading is measured in a euchromatic or heterochromatic environment. I think it is an important point and the data need to be separated accordingly.

This is a good point and something we should have clarified in the manuscript. In the DiVA cell system, we only investigate 53BP1 dynamics at DNA double strand breaks within the euchromatin environment since AsiSI does not induce DSBs in heterochromatin (Massip et al. 2010 Cell Cycle). We now address this issue in the manuscript

text (page 6) and confirm DiVA DSBs are indeed within euchromatin by a Pearson correlation analysis of heterochromatin protein 1 alpha (HP1 α -eGFP) localisation with γ H2AX immunofluorescence (**Fig. S2b**).

3. Why wasn't the 53BP1 homodimerization measured with fluorescent protein-protein interaction visualization or in fixed cells the proximity ligation assay would be good tool to demonstrate that 53BP1 mostly exist in an already dimerized form.

We thank Reviewer 1 for this comment and agree that we should have demonstrated 53BP1 dimerisation in live cells under basal conditions by an independent methodology to Number and Brightness (NB) analysis that is not based on fluorescence fluctuation spectroscopy (FFS). To address issue, we now perform fluorescence anisotropy imaging microscopy (FAIM) on eGFP-53BP1 (new **Fig. 1**) and from detection of homo-Förster resonance energy transfer (FRET) (an alternative read out of protein-protein interaction explained in a new **Fig. S1**) confirm that 53BP1 exists as a dimer throughout the nucleoplasm of DiVA cells before DSB induction (new **Fig. 1m**) and importantly, this population of 53BP1 in a complex spatially redistributes to DiVA DSB sites after addition of 4OHT (**Fig. 2g-h**). We did not measure 53BP1 homodimerisation in fixed cells because it is known from co-immunoprecipitation experiments that 53BP1 dimers exist in the nucleoplasm independent of DNA damage signalling (Adams et al. 2005 Cell Cycle) and we were interested in 53BP1 dimer dynamics (transient and or stable) as well as how they respond to DSB induction in time.

4. The authors use conventional microscopy with an advanced processing method to clarify their findings. The resolution of a conventional confocal microscopic method is approximately 200-300nm. This should be discussed in the text. Additionally, a recent report from Claudia and Jiri Lukas lab used showed mechanistical data on 53BP1 loading by using 3D-SIM. This has to be discussed in the manuscript. Additionally, I also suggest applying quantitative super resolution microscopy at least to show that the 53BP1 is associated to DSB in dimer form (Varga 2019 Nanoscale).

Yes Reviewer 1 is correct, both Number and Brightness (NB) and cross pair correlation function (pCF) analysis are performed on microscopy data that is diffraction limited and we access single molecule information from observation of fluctuations in fluorescence intensity due to eGFP-53BP1 molecules diffusing in and out of pixels with a radial axis of ~ 260 nm (Priest et al. 2019. Biochem. Soc. Trans). As suggested by Reviewer 1 we now discuss this point more extensively in the Methods section when describing our FFS data acquisition (page 18) and also now highlight this fact in the results section when referring to **Fig. 1b** for NB (page 4) and **Fig. 3j** for cross pCF (page 8). The recent 3D-SIM study from Claudia and Jiri Lukas that demonstrated 53BP1 foci exhibit sub-domains spanning 60-180 nm is very interesting and we now reference this important result (Ochs et al. 2019 Nature) alongside the dSTORM study (Varga et al. 2019 Nanoscale) in the discussion (page 15) commenting that it will be critical in future studies for us and others to place the dynamic picture of 53BP1 self-association that we detect in the context of a super-resolved image of 53BP1 foci formation. We make this comment because obtaining the stoichiometry of a fluorescent protein via use of super-resolution microscopy methods such as dSTORM remains an outstanding challenge due to the photophysical properties of the probes used still not being well enough understood to do molecular counting (Feher et al. 2019. Current Opinion Chem Biol.).

5. In the manuscript the pre-existing endogenous 53BP' foci must be addressed or at least discussed (Figure 1e, Figure 2e, Figure 3a and e).

A background of spontaneous DNA DSBs is reported in DiVA cells (Massip 2010 Cell cycle) and we now present quantification of eGFP-53BP1 and γ H2AX immunofluorescence (new **Fig. S2**) that demonstrates the pre-existing endogenous 53BP1 foci observed in the main text figures before 4OHT treatment correlate in number (**Fig. S2d**) with the reported levels of background DNA DSBs (**Fig. S2a** and Massip 2010 Cell Cycle).

6. Figure 1g and h the authors conclude that the amount of monomer and dimer forms of 53BP1 are decreasing while oligomerisation could be observed (This result suggests that upon DSB induction pre-formed 53BP1 dimers in the nucleoplasm are recruited to DSB lesions and upon arrival assemble into higher order oligomers.) For me the data represented on figure 1h show the opposite. Additionally, Figure 1i is not discussed in the text.

The data presented in Fig. 1g (which is now presented as a stacked bar graph in **Fig. 2e**) shows a decrease in the fraction of 53BP1 dimer detected in the nucleoplasm (green profile) as a function of time after 4OHT treatment (53BP1 monomer is presented as the teal profile). This loss of 53BP1 dimer from the nucleoplasm was found to be significantly different from basal conditions 60 min after 4OHT treatment (box and whisker plot **Fig. 2e**) - a result we have now confirmed via homo-FRET (**Fig. 2g**). The data presented in Fig. 1i (which is now presented as a stacked bar graph in **Fig. 2f**) is now discussed in the results section (page 6) and it demonstrates that 53BP1 is assembled into a steady state population of higher order oligomer (i.e. approximately 50 % of 53BP1 foci composition) at the increasingly numerous DSB foci (box and whisker plot **Fig. 2f**).

Minor points:

1. On figure 2 4, 7- and 12-pixel distances are shown. I think it would be better if they would give the distance in nm resolution.

This is a good point and in response we now provide the nanometre distance that corresponds to a cross pair correlation function performed at $\delta r = 4, 7$ and 12 pixels in the figure caption of what is now **Fig. 3** as well as in the results text discussing this analysis (page 8). We have left use of the terms pCF4, pCF7 and pCF12 in the **Fig. 3** panels because this indicates the calculation presented and is consistent with the nomenclature we have used in previous publications demonstrating the pair correlation method (e.g. Hinde et al. 2016 Nature Communications).

2. On figure 3 since the 53BP1 foci formation is affected the authors need to use control to show that the DSB induction is good (ATM-mCherry for example).

We thank Reviewer 1 for this comment and agree that we should have demonstrated that the 53BP1 foci formed by the different 53BP1 mutants unable to bind the DSB histone code are indeed recruited to DiVA DSBs. To address this issue, we have now performed γ H2AX immunofluorescence 60 min after 4OHT treatment and co-localised this signal with the different eGFP tagged 53BP1 mutants (new **Fig. 4a-d**). Quantification of the number of eGFP-53BP1^{D1521R} and eGFP-53BP1^{L1619A} foci formed across multiple cells found that there are significantly less in number than wild type 53BP1 and they are significantly less enriched (new **Fig. 4e**).

3. In order to show that H4K20me and H2aK15ub indeed affects the 53BP1 loading they should perform shRNA knockdown of these PTM writers (SUV4-20h1/2 and RNF8-RNF168 double KD) to validate their results.

We thank Reviewer 1 for this important suggestion. We have now verified that 53BP1 dimer interaction with H2AK15ub and H4K20me2 is critical for 53BP1 dimer loading onto a DNA DSB and subsequent retention at these nuclear locations (page 12) by extension of a two channel cross pair correlation function analysis to 53BP1^{L1619A} and 53BP1^{D1521R} dimer transport (new **Fig. 5**) and analysis of wild type 53BP1 dimer transport upon siRNA knock down of RNF8/RNF168 and chemical inhibition of SUV4-20h1/2, respectively (new **Fig. S7**).

4. In the introduction the authors say that 53BP1 is the heart of DDR. I think it is an overstatement and has to be removed. I agree that this is an important player but not the middle of DDR.

In the introduction (page 2) we now refer to 53BP1 as a key player in the DNA damage response.

5. The following statement also need to be modified: 53BP1 is rapidly recruited to DSBs. Since 53BP1 is considered as not an early factor this sentence needs to be re-phrased.

In the introduction (page 2) we have removed the word rapidly.

6. on page 5 last chapter it is written: the population of 53BP1 molecules being tracked was lost. What does it mean?

This sentence was intended to mean that when we perform cross pair correlation function (pCF) analysis at a distance of $\delta r = 12$ pixels we do not detect 53BP1 dimer translocation with significant efficiency (in other words 0 % of the molecules at the starting pixel arrive at the cross correlated pixel). However, we now see that the original statement queried was ambiguous and thus we have changed the results text to describe the cross pCF profiles in terms of efficiency (page 8), which we now define in the Methods section (page 21) and new **Fig. S5**.

Reviewer # 2 Remarks to the Author):

The manuscript entitled “Stepwise 53BP1 foci assembly in response to DNA double strand break” by Lou and colleagues reports that 53BP1 foci are assembled at sites of DSBs by recruitment of pre-formed 53BP1 dimers via a H2AK15ub dependent mechanism, and upon arrival, H4K20me2 is critical for immobilization, while H2AK15ub stimulates formation of higher order oligomers that lead to a mature DSB repair focus. This conclusion is based entirely on the FFS technique and would be greatly strengthened, if similar conclusions could be reached by alternative methods. The authors confirm previous reports that binding of 53BP1 to H4K20me2 is required for recruitment of 53BP1 to foci, but surprisingly observe recruitment of the 53BP1-L1619A mutant, which is defective in H2AK15ub binding, to foci, which is contradictory to the literature. Also, the manuscript ignores reports that 53BP1 also interacts with phospho-H2AX-S139 (Kleiner et al., 2015), which limits the scope of the work. Given these concerns and additional issues listed below, I recommend that this work would be better suited for a specialist journal where space would permit proper explanation of the fluorescence fluctuation spectroscopy technique and addition of relevant control and complementary experiments.

We thank Reviewer 2 for their overall feedback that motivated us to conduct new experiments and significantly strengthen our manuscript. In particular, in response to the above comments we have now: (1) performed homo-FRET measurements (new **Fig. S1**) to confirm our fluorescence fluctuation spectroscopy (FFS) based conclusions of eGFP-53BP1 dimer-oligomer dynamics under basal conditions (new **Fig. 1**) versus after DSB induction (**Fig. 2g-h**), (2) verified our original finding that H2AK15ub mediates 53BP1 dimer loading onto a DSB while H4K20me2 is important for 53BP1 retention at these nuclear locations by siRNA knockdown of RNF8/RNF168 and chemical inhibition of SUV4-20h1/2 (new **Fig. S7**) (which collectively is in agreement with the literature because it demonstrates *both* histone marks are required for 53BP1 accumulation at a DSB), (3) extended our FFS analysis to investigate the facilitatory role of γ H2AX (Kleiner et al. 2015) on 53BP1 dynamics alongside H2AK15ub and H4K20me2 (new **Fig. 4** and **Fig. 5**) and (4) expanded our explanation of both Number and Brightness (NB) (new **Fig. 1**) and cross pair correlation function (pCF) (**Fig. 3** and new **Fig. S5**) analyses in the results text and Methods section. Below we also detail how we have addressed all major and minor concerns.

Major concerns:

1. The authors do not report which fraction of the 53BP1 molecules in the cell are GFP-tagged. Western blotting should be used to confirm the level of endogenous 53BP1 compared to the GFP-53BP1 transgene.

In a new supplementary figure (**Fig. S4**) we have now performed immunofluorescence against 53BP1 in DiVA cells transiently transfected with eGFP-53BP1 60 min after 4OHT treatment and quantified the signal of endogenous 53BP1 with eGFP-53BP1 (**Fig. S4a**). As a result of this analysis we find transfected eGFP-53BP1 expression to be 1.5-fold higher than endogenous 53BP1, and importantly, our NB quantification of eGFP-53BP1 oligomerisation to be maintained upon siRNA knock down of endogenous 53BP1 (**Fig. S4b-c**).

2. The calibration of GFP-53BP1 to oligomeric state is not convincing. Foci of different oligomeric states should have relative fluorescence intensities of 1, 2, 3, 4, 5, etc, if the authors are truly detecting single molecules. The authors need to introduce a control where the oligomeric state is known with certainty e.g. a chromatin-bound protein fused to 1, 2, 3 and 4 molecules of GFP.

We thank Reviewer 2 for this comment, which has prompted us to assemble a new main text figure (**Fig. 1**) dedicated to explaining the key principle behind Number and Brightness (NB) analysis (**Fig. 1a-b**) and validation that our NB analysis detects bona fide 53BP1 oligomerisation (**Fig. 1c-m**). This figure now includes: (1) calibration of eGFP-53BP1 dimer-oligomer brightness windows (**Fig. 1c-f**), (2) application of the calibrated brightness windows to wild type eGFP-53BP1 versus 53BP1 mutants (Zgheib 2009 Mol Cell Biol) that exhibit reduced dimerization (eGFP-53BP1^{YY1258,1259AA}, negative control) versus constitutive oligomerisation (eGFP-GCA-53BP1, positive control) (**Fig. 1g-j**), and (3) independent verification that our NB analysis workflow reports 53BP1 dimer-oligomer formation by fluorescence anisotropy imaging microscopy of homo-FRET (**Fig. 1k-m**) - an alternative read out of protein-protein interaction that is not based on fluorescence fluctuation spectroscopy.

Importantly, foci of different oligomeric states will *not* exhibit a relative fluorescence intensity because NB analysis retrieves eGFP-53BP1's oligomeric state from *fluctuations* in fluorescence intensity due to eGFP-53BP1 molecules diffusing in and out of a diffraction limited pixel (Digman et al. 2009. Biophys J and Hinde et al. 2016. Nature Communications). The *mean* intensity in each pixel is in part the result of an immobile fraction of eGFP-53BP1 molecules that do not contribute to the fluorescence *fluctuation*, and therefore, the fluorescence intensity of a pixel is not indicative of the eGFP-53BP1 brightness that is recovered at that location. This is an aspect of NB we should have emphasised in the manuscript and we now do so in the Methods section (page 18-19).

3. Introduction: the current knowledge on 53BP1 oligomeric states (monomers, dimers, tetramers) should be reviewed in more detail to put the study in context.

Yes, we agree with Reviewer 2 and in response we have modified our introduction to review in more detail the current knowledge on 53BP1 dimerisation and oligomerisation (page 2-3).

4. Fig. 1f-g: Have the images been adjusted for photobleaching? It looks like the overall fluorescence decreases with time.

The intensity images in Fig. 1e (now **Fig. 2c**) were autoscaled and they are now presented on a set intensity axis of 0-400 arbitrary units. The brightness images in Fig. 1f (now **Fig. 2d**) are not adjusted for any photobleaching that may occur *between* acquisitions, but as described in the Methods section (page 19), a moving average is applied across the $n = 100$ intensity frames from which each brightness map is derived, to eliminate any artefact from photobleaching *during* an acquisition. The change observed in the intensity versus brightness distributions presented in Fig. 1g (now **Fig. S3a**) is largely due to the spatial redistribution of eGFP-53BP1 molecules from the nucleoplasm to DSB foci and self-association into higher order oligomers. Any photobleaching that may occur between NB acquisitions is eliminated from our quantification of 53BP1 dimer-oligomer formation before versus after 4OHT treatment (**Fig. 2e-f**) since this data was acquired from independent cells at the different time points (representative examples are now presented in **Fig. S3b-c** and they recapitulate the spatiotemporal redistribution in 53BP1 oligomerisation observed in the single cell presented in **Fig. 2c-d**).

5. Fig. 1f-g: The indication of oligomers as tetramers is misleading. If I understand the data correctly, there brighter simply contains a greater number of 53BP1 molecules above a certain threshold. The time-lapse quantification looks more like immobilization of diffuse nuclear 53BP1 at DSBs. What is the number of 53BP1 molecules in the brighter foci?

We obtain the brightness of a molecule from *fluctuations* in fluorescence intensity not the *mean* fluorescence intensity and so a pixel containing a greater number of fluorescent molecules does not necessarily recover a higher brightness value. For example, if we have 4 monomers diffuse in and out of a pixel versus 1 tetramer, although they have an equivalent mean intensity, the fluctuation in fluorescence intensity will be greater in the case of the tetramer. The intensity versus brightness scatter plots presented in Fig. 1f-g (now **Fig. S3a** since we now define the brightness palette in **Fig. 1e**) have the oligomer cursor (red) centred at the brightness of a tetramer, however, we agree that we should simply refer to this species as an oligomer, since the variation in brightness detected by this cursor extends from a trimer to a pentamer. We therefore have changed all reference to tetramers in the manuscript text to oligomers. The time series of brightness maps presented in **Fig. 1f-g** (now **Fig. 2c-d**) show 53BP1 oligomer formation (red pixels) (**Fig. 2d**) at 53BP1 foci (**Fig. 2c**) and indeed we later show by cross pair correlation function (pCF) analysis (**Fig. 4**) that 53BP1 accumulation at these nuclear locations results from an immobilisation of 53BP1 dimers at these locations. However, in order for NB to detect the 53BP1 oligomer population at a D_{IV}A DSB it must be immobilised by a dynamic event such as a binding interaction, since like all fluorescence fluctuation-based methods, NB does not detect a truly immobile fraction. In what is now **Fig. 2c-d** the mean number of *mobile* particles detected per pixel in 53BP1 foci 60 min after 4OHT treatment is 11 ± 5 .

6. Fig 1h: N = 10 cells. This is a very low number of cells. How many biological replicates were analysed?

In Fig 1h (now **Fig. 2e-f**) N = 10 cells was the result of N = 5 cells acquired across two biological replicates. We have now acquired Number and Brightness (NB) data before and after 4OHT treatment in N = 5 cells across six biological replicates (i.e. total N = 30 cells) and this data is presented in **Fig. 1k-l** (N = 10 cells, two biological replicates), **Fig. 2e-f** (N = 10 cells, two biological replicates) and **Fig. 4i-j** (N = 10 cells, two biological replicates).

In each case we observe a significant loss of 53BP1 dimer from the nucleoplasm 60 min after 4OHT treatment and 53BP1 oligomer formation at the significantly increased number of 53BP1 foci. This data is presented in box and whisker plots that show the minimum, the maximum, the sample median, and the first and third quartiles. Significance between data sets was assessed via an unpaired t test.

7. Fig 1i: The fraction of pixels in foci does not reflect the example in Fig 1f, where it looks like the number of red pixels is similar at all time points.

The fraction of pixels in 53BP1 foci in the example presented in Fig 1f (now **Fig. 2c-d**) is 2 % before and 0 min after 4OHT treatment, and this foci fraction increases to 4 % by 30 min to 60 min after 4OHT treatment. The number of pixels identified as being oligomeric (red pixels) within the foci fraction at the different time points is 237, 372, 692 and 619, respectively. Both of these trends reflect the quantification presented in what is now **Fig. 2f** and which shows 53BP1 oligomer formation at an increasing number of 53BP1 foci following 4OHT treatment.

8. Are the eGFP-53BP1 and mKate2-53BP1 constructs expressed at equal levels. Western blots should be provided.

In the DiVA cells selected for cross pair correlation function (pCF) analysis the expression is approximately equivalent - specifically the number of mKate2-53BP1 molecules present is 1.3-fold higher than eGFP-53BP1. We know this from the pCF0 analysis performed in **Fig. 3h** (was Fig. 2h) that quantifies the fraction of 53BP1 dimer present in the nucleoplasm (yellow profile) from the total number of molecules in the eGFP-53BP1 channel (green profile) versus the mKate2-53BP1 channel (red profile). As described in the methods section (page 20-21) the total number of molecules present in each channel is extracted from the pCF0 analysis, which is an autocorrelation function (ACF) and a well-established read out of particle concentration at $\tau = 0$ (i.e. the y-intercept (termed $G(0)$) reports the number of molecules in each pixel along the line scan according to $N_{\text{molecules}} = \gamma / G(0)$ where γ describes the shape of a 1-photon point spread function) (Priest et al. 2019. Biochem Soc. Trans.)

9. The co-expression of eGFP-53BP1 and mKate2-53BP1 should reveal monomers that are either green or red, and dimers that are either half green and half red or entirely green or red (assuming no endogenous 53BP1 and equal amounts of eGFP-53BP1 and mKate2-53BP1 are expressed). Can this be observed?

Yes Reviewer 2 is correct, a two-channel line scan acquired in DiVA cells co-expressing eGFP-53BP1 (CH1) and mKate2-53BP1 (CH2) enables detection of: (1) eGFP-53BP1 monomer and homo-dimer mobility by pair correlation function analysis of CH1, (2) mKate2-53BP1 monomer and homo-dimer mobility by pair correlation analysis of CH2, and finally, (3) eGFP-53BP1 and mKate2-53BP1 hetero-dimer mobility by cross pair correlation analysis of CH1 with CH2. A schematic of this capacity is depicted in what is now **Fig. 3a-d** (was Fig. 2a-d). We thus use the cross pair correlation function analysis to assess 53BP1 dimer mobility since monomers are absent.

10. Fig 3: very few cells are analysed and apparently only 1 biological replica? What is the variation in multiple biological replicates?

We have now acquired Number and Brightness (NB) data of wild type 53BP1 in $N = 5$ cells across 6 biological replicates (i.e. $N = 30$) and NB data of the different 53BP1 mutants (D1521R, L1619A and K1814M) in $N = 5$ cells across two biological replicates (i.e. $N = 10$). In each case the results before and after 4OHT treatment were consistent. This data is presented in box and whisker plots that show the minimum, the maximum, the sample median, and the first and third quartiles. Significance between data sets was assessed via an unpaired t test. We have also now performed cross pair correlation function (pCF) analysis on wild type 53BP1 in $N = 5$ cells across 3 biological replicates (i.e. $N = 15$) and cross pCF analysis on the different 53BP1 mutants (D1521R, L1619A and K1814M) in $N = 5$ cells across two biological replicates (i.e. $N = 10$). In each case the results before and after 4OHT treatment were consistent. Pair correlation profiles are shaded with the standard error of the mean.

11. Fig 4d-f: The eGFP-53BP1-L1619A foci observed after DSB-induction have a different morphological appearance than WT foci. The authors should confirm that these foci colocalized with gamma-H2AX and are not inclusion bodies due to the L1619A mutation. Notably, other labs have reported that 53BP1-L1619A is defective for recruitment to DSB-induced foci (Fradet-Turcotte et al., 2013).

We thank Reviewer 1 for this comment and agree that we should have demonstrated that the 53BP1 foci formed by eGFP-53BP1^{L1619A} (as well as eGFP-53BP1^{D1521R} and eGFP-53BP1^{K1814M}) are indeed recruited to DiVA DSBs. To address this issue, we have now performed γ H2AX immunofluorescence 60 min after 4OHT treatment and co-localised this signal with the different eGFP tagged 53BP1 mutants (new **Fig. 4a-d**). As can be seen in **Fig. 4b** we find that the majority of eGFP-53BP1^{L1619A} foci do colocalise with γ H2AX foci and therefore are located at DiVA DSBs, but in agreement with the literature that reports defective eGFP-53BP1^{L1619A} recruitment, these foci are significantly rarer and less enriched than wild type 53BP1 foci (as quantified in **Fig. 4e**).

12. The assessment of monomeric and dimeric forms of 53BP1 should be complemented by alternative methods e.g. FLIM-FRET.

We thank Reviewer 1 for this comment and agree that we should have demonstrated 53BP1 oligomerisation in live cells under basal conditions by an independent methodology to Number and Brightness (NB) analysis that is not based on fluorescence fluctuation spectroscopy (FFS). Thus in response we now perform fluorescence anisotropy imaging microscopy (FAIM) on eGFP-53BP1 (new **Fig. 1**) and from detection of homo-Förster resonance energy transfer (FRET) (an alternative read out of protein-protein interaction explained in a new **Fig. S1**) confirm that a dimeric population of 53BP1 exists throughout the nucleoplasm of DiVA cells before DSB induction (new **Fig. 1m**) and importantly this population of 53BP1 dimer spatially redistributes to DiVA DSB sites after addition of 4OHT (new **Fig. 2g-h**).

13. The authors propose that dimers and monomers have different on/off rates at foci. This should be supported by FRAP analysis.

We thank Reviewer 2 for this comment, however, to our knowledge FRAP is not able to differentiate monomers from dimers. Also, the directionality of the cross-pair correlation function is required to differentiate the arrival time *onto* the DSB from the arrival time *off* this structure (Hinde et al. 2016. Nature Communications).

Minor comments:

1. Fig S1a: There appears to be no gammaH2AX foci in the untreated sample and also no background in these cells. This is unlike what is normally reported in the literature, where a background of spontaneous foci are normally observed. Were the images subjected to the same contrast enhancement as the 4OHT induced cells?

Yes, the same contrast was applied to the intensity images presented in Fig. S1a (now **Fig. S2a**) and this is likely why it is difficult to see spontaneous foci in the image taken before 4OHT treatment. However as presented in the bar graph of Fig. S1a (now **Fig. S2a**) quantification of this data demonstrates that on average we detect 2 ± 1 spontaneous foci before 4OHT treatment (which is in agreement with Massip 2010 Cell Cycle).

2. Page 3: Fig 1e does not appear to show 15 min intervals according to the labels (-10, 0, 10, 30 min)?

Yes Reviewer 2 is correct. We apologise for this error and have fixed the image labels in what is now **Fig. 2c-d** (was Fig. 1e) to correctly report the time our NB data acquisitions were acquired with respect to 4OHT treatment (i.e. - 10 min (before), 0 min, 30 min and 60 min).

3. Page 12: please reference the “previously published papers”.

We apologise for this error and we now include the references to the previously published papers (page 18-20).

4. Page 5: I don't think the data justifies 4 significant digits (27.62%).

Yes Reviewer 2 is correct we have adjusted the calculated fraction of 53BP1 dimer to 26 ± 4 % (page 4).

5. Page 7: it has not been formally shown that 53BP1 dimers “self-associate into higher order oligomers”. It might be that multiple dimers bind to individual nucleosomes within a focus.

Number and Brightness (NB) analysis *does* formally show that 53BP1 self-associates into higher order oligomers since the fluorescence *fluctuation* of an eGFP-53BP1 oligomer diffusing through the point spread function of a pixel in our frame scan is inherently different from the fluorescence *fluctuation* two eGFP-53BP1 dimers would give rise to when diffusing through a point spread function of a pixel in our frame scan. This difference is now

more clearly explained in the results text (page 4) when describing *Fig. 1a-b* that is dedicated to establishing the principle behind NB analysis before demonstration of this method on eGFP-53BP1.

6. Page 7: what is the evidence that “under this condition only 53BP1 dimers were found to be tracked with respect to DSB foci”?

The evidence for tracking 53BP1 dimers at a cross pair correlation function distance of $\delta r = 7$ (i.e. pCF7) is based on a comparison of the cross pCF7 profile amplitude with the amplitude of the individual channel pCF7 profiles (i.e. eGFP-53BP1 and mKate2-53BP1). The evidence for cross pCF7 being optimal for tracking 53BP1 dimers with respect to a DSB is based on the fact that $\delta r = 4$ (pCF4) was weighted toward local mobility within a pixel while $\delta r = 12$ (pCF12) was too large of a distance to detect 53BP1 dimer translocation with significant efficiency. This reasoning is now described in more details in the methods section (page 20-21).

References:

Fradet-Turcotte, A., Canny, M.D., Escribano-Diaz, C., Orthwein, A., Leung, C.C., Huang, H., Landry, M.C., Kitevski-LeBlanc, J., Noordermeer, S.M., Sicheri, F., et al. (2013). 53BP1 is a reader of the DNA-damage-induced H2A Lys 15 ubiquitin mark. *Nature (London)* 499, 50-54.
Kleiner, R.E., Verma, P., Molloy, K.R., Chait, B.T., and Kapoor, T.M. (2015). Chemical proteomics reveals a gammaH2AX-53BP1 interaction in the DNA damage response. *Nat Chem Biol* 11, 807-814.

Reviewer #3 (Remarks to the Author):

In this manuscript, Lou et al. analyze in living cells the oligomeric state of 53BP1 in and out DNA repair foci using fluorescence fluctuations methods. Based on N&B and pair correlation data, the authors propose that, outside repair foci, 53BP1 mainly exists as dimer, these dimers thus forming higher oligomers upon relocation to repair foci. They also analyze the impact on the 53BP1 oligomerization status of two mutations impairing binding to specific histone marks and propose that H4K20me2 is important for immobilisation of 53BP1 dimers at repair foci while H2AK15ub stimulates formation of higher order oligomers.

We thank Reviewer 3 for their insightful feedback and in response we have now acquired new experimental data to address the major concerns raised below, as well as expanded explanation of the methods employed. Incorporation of this data and information into a majorly revised manuscript has significantly strengthened our biological conclusions and the reproducibility of our technological approach. Highlights based on Reviewer 3's suggestions include: (1) use of siRNA to confirm that the DSB dependent 53BP1 dimer-oligomer dynamics we detect by FFS are biologically relevant and maintained upon knocking down endogenous 53BP1 (new *Fig. S4*), (2) colocalization of γ H2AX immunofluorescence with the eGFP labelled 53BP1 mutant foci to confirm that they are located at DivA DSBs (new *Fig. 4a-d*) and (3) extension of the cross-pair correlation function analysis to eGFP and mKate2 tagged 53BP1 mutants unable to bind the DSB histone code (new *Fig. 5*).

The method presented in this manuscript is interesting and potentially applicable to other biological questions since it allows to assess directly in living cells protein oligomeric state as well as dynamic binding to DNA repair foci. The biological insights which are presented are valuable although relatively limited. In addition, I believe that additional experiments are necessary to more convincingly support the conclusions made by the authors. I would advise the authors to address the following points :

1) The experiments are currently performed using cells over-expressing fluorescently tagged 53BP1. Thus, there is a possibility that 53BP1 dimers are composed of a tagged molecule as well as an untagged one, which would be detected as a monomer by N&B. The coexistence of these two versions of 53BP1 in the cells makes the interpretation of the present results somehow difficult. Either the size of the oligomers is underestimated by the presence of significant amount of untagged 53BP1 or the amount of endogeneous 53BP1 is considered as neglectable compared to the quantity of tagged 53BP1 molecules. However, this last possibly supposes high levels of overexpression of GFP-53BP1, thus questioning biological relevance. I think that it would be important to perform the experiments presented in this manuscript using cells knocked-down or knocked-out for endogeneous 53BP1. In addition, it would also be useful to control that the expression of GFP-53BP1 in

these cells is comparable to endogeneous level to avoid potential artifactual oligomerisation of GFP-53BP1 due to too high expression.

We thank Reviewer 3 for this comment and in response we have now generated a new supplementary figure (**Fig. S4**) where we have now performed immunofluorescence against 53BP1 in DiVA cells transiently transfected with eGFP-53BP1 60 min after 4OHT treatment and quantified the signal of endogenous 53BP1 with eGFP-53BP1 (**Fig. S4a**). As a result of this analysis we find transfected eGFP-53BP1 expression to be 1.5-fold higher than endogenous 53BP1, and importantly, our NB quantification of eGFP-53BP1 oligomerisation to be maintained upon siRNA knock down of endogenous 53BP1 (**Fig. S4b-c**).

2) Fluorescence fluctuation methods usually require relatively low local concentration of molecules to be able to measure fluorescence fluctuations of sufficient amplitudes. This tends to question the ability to obtain accurate N&B results at bright spots such as DSB repair foci. The authors should demonstrate that they can perform N&B analysis at such high concentration of molecules. For example, they could perform N&B in nuclei with fluorescent spots obtained by the accumulation of tagged Lac repressor at Lac operator arrays. Since Lac repressor are known form tetramers (see e.g. Friedman et al., Science, 1995, doi: 10.1126/science.7792597) this could be used as a control to validate N&B analysis.

It is true that Number and Brightness (NB) like all fluorescence fluctuation-based methods requires a relatively low local concentration of *moving* molecules in order to detect a fluctuation in fluorescence intensity. However, in the case of the 53BP1 foci - that are more intense than the nucleoplasm – given that we do detect a *fluctuation* in fluorescence intensity that enables calculation of a variance, a significant portion of the signal from these locations likely originates from an immobile fraction (on the timescale of our experiment). If the concentration of *moving* 53BP1 molecules at these nuclear locations was too high or there was no *moving* particles, then we would not detect a variance in fluorescence intensity (Digman et al. 2008. Biophys J.) Nonetheless, we agree that we should have more thoroughly validated the NB analysis and to address this issue we have now assembled a new **Fig. 1** where we: (1) measure 53BP1 mutants (Zgheib 2009 Mol Cell Biol) that exhibit reduced dimerization (eGFP-53BP1^{YY1258,1259AA}, negative control) versus constitutive oligomerisation (eGFP-GCA-53BP1, positive control) and (2) confirm 53BP1 protein-protein interaction is promoted at 53BP1 foci via homo-FRET.

3) For the two mutants of 53BP1 that do not strongly accumulate at DNA repair foci, it would be important to co-express a second marker allowing to localize the DSBs. This would allow to make sure that pCF analysis is indeed performed along lines crossing DSBs.

We thank Reviewer 3 for this comment and agree that we should have demonstrated that the 53BP1 foci formed by the different 53BP1 mutants unable to bind the DSB histone code are indeed recruited to DiVA DSBs. To address this issue, we have now performed γ H2AX immunofluorescence 60 min after 4OHT treatment and co-localised this signal with the different eGFP tagged 53BP1 mutants (new **Fig. 4a-d**). Also, quantification of the number of eGFP-53BP1^{D1521R} and eGFP-53BP1^{L1619A} foci formed across multiple cells found that there are significantly less in number than wild type 53BP1 (new **Fig. 4e**).

4) For the two mutants of 53BP1, the authors compared single-color pCF curves with the results obtained by cross pCF for the wild-type 53BP1 (Fig 3j-k Fig 4j-k). It seems inappropriate to compare results obtained with two different approaches, one allowing to specifically look at dimers (cross pCF), while the other one probes both monomers and oligomers (single-color pCF). I would suggest that the authors perform cross pCF for the two 53BP1 mutants as they did for the wild-type version of 53BP1. Also, it would be useful to show single-color pCF for wild-type 53BP1 leaving the repair foci. Currently, the authors show by cross pCF that 53BP1 do not leave the foci as dimer but it would be interesting to know whether they could leave the foci as monomers.

This is a very good point and we definitely agree with Reviewer 3 that we should have performed cross pair correlation function (pCF) analysis on the different 53BP1 mutants investigated. To address this issue, we have now generated eGFP and mKate2 labelled constructs of 53BP1^{D1521R} and 53BP1^{L1619A} and acquired two channel line scan FFS data in DiVA cells co-expressing these constructs. From cross pair correlation function analysis of this data (presented in new **Fig. 5**) we find in agreement with our original analysis that H4K20me2 is important for 53BP1 dimer retention at the DSB while H2AK15ub is important for 53BP1 dimer loading onto the DSB.

With respect to pCF analysis of total 53BP1 transport off a DNA DSB, as shown in **Rebuttal Fig. 1b** (right) we find that pCF7 analysis of eGFP-53BP1 in CH1 (green profile) detects a small fraction of molecules leaving the DSB on a fast time scale that is not present in the cross pCF7 analysis that only detects eGFP-53BP1 in complex with mKate2-53BP1 (yellow). This fraction of molecules likely represents the 53BP1 monomer population.

Rebuttal Fig. 1: pCF7 versus cross pCF7 analysis of 53BP1 transport onto a DNA DSB (a) versus off this structure (b).

5) In several instances in the manuscript, the authors give very few details regarding some data processing steps. For example, this is the case for the bleedthrough correction (Fig S3) or the calculation of the fraction of dimers (Fig 2i). It would be important to explain better how these data processing steps are performed.

We thank Reviewer 3 for this comment and agree that we should have provided more detail on how both the Number and Brightness (NB) and cross pair correlation function (pCF) analysis was performed and quantified in our main text figures. In the case of NB, to address this comment, we have extended the methods section (page 18-19) to detail how we use the assigned brightness windows (based on the eGFP calibration presented in **Fig. 1**) to quantify the fraction of 53BP1 dimer or oligomer in what is now **Fig. 2** and **Fig. 4**. In the case of cross pCF, to address this comment, we have extended the methods section (page 20-21) and prepared a new supplementary figure (**Fig. S5**) that details: (1) how we now calculate the fraction of 53BP1 dimer present in the nucleoplasm via cross pCF0 analysis (essentially an autocorrelation function) and then correct this fraction for spectral bleed through (**Fig. S5a**) and (2) how we now calculate the arrival time and fraction (i.e. efficiency) of 53BP1 dimer that translocates to another spatial location via cross pCF4, pCF7 and pCF12 analysis (**Fig. S5b-c**).

6) The current version of the discussion is relatively short. I would find interesting to discuss more extensively the biological implications of the results presented in this manuscript. More specifically, the authors should discuss their results in the context of the recent publication by Kilic et al. (Kilic et al., EMBO J, 2019) which shows that 53BP1 displays the propensity to undergo phase separation in the nucleoplasm.

We thank Reviewer 3 for this suggestion. We now discuss the biological implications of our findings in more detail (page 14-15) and in particular we discuss our findings with respect to the results of Kilic et al. (page 15).

REVIEWERS' COMMENTS:

Reviewer #1 (Remarks to the Author):

The research paper entitled "Stepwise 53BP1 foci assembly in response to DNA double strand break." where the homodimerization process of the 53BP1 following DSB induction is examined. In the revised version of the manuscript significant new data have been provided and the authors addressed all of my questions accordingly. The current form of the manuscript, in my opinion is suitable for publication in Nature communication so I recommend it for acceptance.

Reviewer #2 (Remarks to the Author):

The revised manuscript entitled "Spatiotemporal dynamics of 53BP1 dimer recruitment to a DNA double strand break" by Lou and colleagues and the response of the authors to the reviewer comments address most of the issues that I raised. However, one of my primary concerns has not been appropriately addressed and this regards the expression level of the eGFP-53BP1 transgene. If the expression of the eGFP-53BP1 transgene is low compared with endogenous 53BP1 then many dimers will be hetero-dimers and hence scored as monomers. The authors show that the percentage of dimers after knockdown of endogenous 53BP1 is virtually unchanged at 20%, indicating that the eGFP-53BP1 transgene is significantly overexpressed compared to endogenous 53BP1. Nevertheless, the authors state on page 6 "exhibiting a low eGFP-53BP1 expression level". The word "low" should be more precisely defined. In the response of the authors to the reviewer's comments they state that the eGFP-53BP1 transgene is overexpressed 1.5 fold, but this is based on Fig. S4a, which I believe is a flawed experiment. I do not understand how a comparison of an immunofluorescence signal intensity with live cell eGFP fluorescence intensity can inform on the relative amount of eGFP-53BP1 and endogenous 53BP1. First, the quantum efficiency of the fluorescent antibody is likely to be different than that of eGFP. Second, the fluorescent antibody will only recognize a subset of molecules in the nucleus and presumably recognize both eGFP-53BP1 and endogenous 53BP1. I suggest that Figure S4a is deleted and the authors conclude based on S4b-c that the eGFP-53BP1 transgene is sufficiently overexpressed that formation of hetero-dimers with endogenous 53BP1 constitute only a minor fraction of the dimers.

Minor correction:

Page 14: change "stabilise" to "stabilize".

Reviewer #3 (Remarks to the Author):

The authors made valuable efforts to address the concerns of the different referees, including my own comments. The new data that are provided significantly strengthen the conclusions of this work. I would be happy to recommend this work for publication provided that the minor remaining points below are addressed :

Figs 2 and 4: Currently, it is written in the legend * $P < 0.5$. I assume that the authors rather meant $P < 0.05$.

Fig S4a : please explain more clearly how was obtained over-expression measurements shown on the bottom barplot. What was exactly measured in the eGFP-53BP1 condition in terms of intensity ?

Fig S5 : The legend of this figure remains difficult to read. Please give letters to each of the different panels to help the reader to navigate through the figure in parallel to reading the figure legend.

Material and Methods : I suppose that the bleed through correction that is applied assumes that green- and red-labeled protein are expressed at similar levels. If it is indeed the case, I think the authors should mention this and say that they can verify that they fulfill this condition by looking at the relative amplitudes of the single-color pCF curves.

Axis ticks are missing in all the graphs showing correlation curves (e.g. Fig 3g,h,ik,l, same for Fig 5...) as well as many other graphs. The author should make sure to solve this issue.

Fig 2f middle panel : I am not sure what is plotted here. Is this the fraction of pixel relative to the whole nucleus or only within the foci ? If it is the latter, why don't the bar plot reach 1 as in the middle graph of panel e. Also, this representation in the middle graph of panel e seems different from the one used in the bottom right graph of Fig S4, thus preventing proper comparison between the two experiments. I would advise the author to try to unify the data representation they use and clarify these points.

In the last paragraph of the results, the authors assess the proportion of retention efficiency (like e.g. in the sentence "the net population of 53BP1 dimer retained at the DSB site is reduced to an efficiency of 10 %, because significant 53BP1 dimer transport off the DSB is enabled"). Can the authors better explain how this retention efficiency is estimated and how much they can assess significant differences for this parameter between the wild-type and mutant 53BP1 ?

Response to Reviewers Comments (NCOMMS-19-39894A-Z)

Reviewer #1 (Remarks to the Author): The research paper entitled "Stepwise 53BP1 foci assembly in response to DNA double strand break." where the homodimerization process of the 53BP1 following DSB induction is examined. In the revised version of the manuscript significant new data have been provided and the authors addressed all of my questions accordingly. The current form of the manuscript, in my opinion is suitable for publication in Nature communication so I recommend it for acceptance.

Reviewer #2 (Remarks to the Author): The revised manuscript entitled "Spatiotemporal dynamics of 53BP1 dimer recruitment to a DNA double strand break" by Lou and colleagues and the response of the authors to the reviewer comments addresses most of the issues that I raised. However, one of my primary concerns has not been appropriately addressed and this regards the expression level of the eGFP-53BP1 transgene. If the expression of the eGFP-53BP1 transgene is low compared with endogenous 53BP1 then many dimers will be hetero-dimers and hence scored as monomers. The authors show that the percentage of dimers after knockdown of endogenous 53BP1 is virtually unchanged at 20%, indicating that the eGFP-53BP1 transgene is significantly overexpressed compared to endogenous 53BP1. Nevertheless, the authors state on page 6 "exhibiting a low eGFP-53BP1 expression level". The word "low" should be more precisely defined. In the response of the authors to the reviewer's comments they state that the eGFP-53BP1 transgene is overexpressed 1.5-fold, but this is based on Fig. S4a, which I believe is a flawed experiment. I do not understand how a comparison of an immunofluorescence signal intensity with live cell eGFP fluorescence intensity can inform on the relative amount of eGFP-53BP1 and endogenous 53BP1. First, the quantum efficiency of the fluorescent antibody is likely to be different than that of eGFP. Second, the fluorescent antibody will only recognize a subset of molecules in the nucleus and presumably recognize both eGFP-53BP1 and endogenous 53BP1. I suggest that Figure S4a is deleted and the authors conclude based on S4b-c that the eGFP-53BP1 transgene is sufficiently overexpressed that formation of hetero-dimers with endogenous 53BP1 constitute only a minor fraction of the dimers. *We thank Reviewer 2 for this comment, as suggested we have now deleted panel A from Supplementary Figure 4 and adjusted the manuscript text.*

Minor correction: Page 14: change "stabiise" to "stabilise". *This error has now been corrected.*

Reviewer #3 (Remarks to the Author): The authors made valuable efforts to address the concerns of the different referees, including my own comments. The new data that are provided significantly strengthen the conclusions of this work. I would be happy to recommend this work for publication provided that the minor remaining points below are addressed:

Figs 2 and 4: Currently, it is written in the legend * $P < 0.5$. I assume that the authors rather meant $P < 0.05$. *We thank Reviewer 3 for picking up this error, it has now been corrected.*

Fig S4a: please explain more clearly what was obtained in the over-expression measurements shown on the bottom bar-plot. What was exactly measured in the eGFP-53BP1 condition in terms of intensity? *This experiment was intended to compare endogenous 53BP1 levels (based on IF signal) with eGFP-53BP1 expression at the single cell level. However, as suggested by Reviewer 2 we have now removed panel A from Supplementary Fig. 4*

Fig S5: The legend of this figure remains difficult to read. Please give letters to each of the different panels to help the reader to navigate through the figure in parallel to reading the figure legend. *We have now given letters to each panel and clarified the figure legend.*

Material and Methods: I suppose that the bleed through correction that is applied assumes that green- and red-labelled protein are expressed at similar levels. If it is indeed the case, I think the authors should mention this and say that they can verify that they fulfill this condition by looking at the relative amplitudes of the single-color pCF curves. *We thank Reviewer 3 for this comment, this is correct, and we have clarified this point in the Methods section.*

Axis ticks: missing in all the graphs showing correlation curves (e.g. Fig 3g,h,ik,l, same for Fig 5...) as well as many other graphs. The author should make sure to solve this issue. *We thank Reviewer 3 for this comment, and we have now added axis ticks to all correlation curves / graphs.*

Fig 2f middle panel : I am not sure what is plotted here. Is this the fraction of pixel relative to the whole nucleus or only within the foci ? If it is the latter, why don't the bar plot reach 1 as in the middle graph of panel e. Also, this representation in the middle graph of panel e seems different from the one used in the bottom right graph of Fig S4, thus preventing proper comparison between the two experiments. I would advise the author to try to unify the data representation they use and clarify these points. *We thank Reviewer 3 for this comment. In response we now plot the fractional contribution of monomer, dimer, and oligomer within the selected foci region of interest in Fig. 2f, so that as suggested, we unify the data representation with Supplementary Fig. 4.*

In the last paragraph of the results: the authors assess the proportion of retention efficiency (like e.g. in the sentence "the net population of 53BP1 dimer retained at the DSB site is reduced to an efficiency of 10 %, because significant 53BP1 dimer transport off the DSB is enabled"). Can the authors better explain how this retention efficiency is estimated and how much they can assess significant differences for this parameter between the wild-type and mutant 53BP1? *We thank Reviewer 3 for this comment. The retention efficiency is the difference in amplitude at the peak delay time between the normalised cross pair correlation profile for 53BP1 dimer arrival onto a DSB versus 53BP1 dimer departure from a DSB. We now clarify this point in the Methods section. In the sentence cited we say 'significant 53BP1 dimer transport off the DSB is enabled' because in this instance we recover a positive cross pair correlation profile for diffusion off the DSB and since we used a normalised cross correlation function, this result must be statistically significant. In terms of assessing significant differences between pair correlation profiles derived for wild type 53BP1 versus mutants we plot the standard error of the mean, however, an alternative approach could be selection of a time scale of interest, recording the pair correlation amplitude at this delay time for a specific transit across multiple cells and application of standard significant tests.*